# LONG-HORIZON VIDEO PREDICTION USING A DYNAMIC LATENT HIERARCHY

## ABSTRACT

The task of video prediction and generation is known to be notoriously difficult, with the research in this area largely limited to short-term predictions. Though plagued with noise and stochasticity, videos consist of features that are organised in a spatiotemporal hierarchy, different features possessing different temporal dynamics. In this paper, we introduce *Dynamic Latent Hierarchy* (DLH) – a deep hierarchical latent model that represents videos as a hierarchy of latent states that evolve over separate and fluid timescales. Each latent state is a mixture distribution with two components, representing the immediate past and the predicted future, causing the model to learn transitions only between sufficiently dissimilar states, while clustering temporally persistent states closer together. Using this unique property, DLH naturally discovers the spatiotemporal structure of a dataset and learns disentangled representations across its hierarchy. We hypothesise that this simplifies the task of modeling temporal dynamics of a video, improves the learning of long-term dependencies, and reduces error accumulation. As evidence, we demonstrate that DLH outperforms state-of-the-art benchmarks in video prediction, is able to better represent stochasticity, as well as to dynamically adjust its hierarchical and temporal structure. Our paper shows, among other things, how progress in representation learning can translate into progress in prediction tasks.

## 1 INTRODUCTION

Video data is considered to be one of the most difficult modalities for generative modelling and prediction, characterised by high levels of noise, complex temporal dynamics, and inherent stochasticity. Even more so, modelling long-term videos poses a significant challenge due to the problem of sequential error accumulation, largely restricting the research in this topic to short-term predictions.

Deep learning has given rise to generative latent-variable models with the capability to learn rich latent representations, allowing to model high-dimensional data by means of more efficient, lower-dimensional states (Kingma & Welling, 2014; Higgins et al., 2022; Vahdat & Kautz, 2020; Rasmus et al., 2015). Here, of particular interest are hierarchical latent models, which possess a higher degree of representational power and expressivity. Employing hierarchies has so far proved to be an effective method for generating high-fidelity visual data, as well as concurrently producing more meaningful and disentangled latent representations in both static (Vahdat & Kautz, 2020) and temporal (Zakharov et al., 2022) datasets.

Unlike images, videos possess a *spatiotemporal* structure, in which a collection of spatial features adhere to the intrinsic temporal dynamics of a dataset – often evolving at different and fluid timescales. For instance, consider a simplistic example shown in Figure 1, in which the features of a video sequence evolve within a strict temporal hierarchy: from the panda continuously changing its position to the background elements being static over the entire duration of the video.

Discovering such a temporal structure in videos complements nicely the research into hierarchical generative models, which have been shown capable of extracting and disentangling features across a hierarchy of latent states. Relying on this notion of inherent spatiotemporal organisation of features, several hierarchical architectures have been proposed to either enforce a generative temporal hierarchy explicitly (Saxena et al., 2021), or discover it in an unsupervised fashion (Kim et al., 2019; Zakharov et al., 2022). In general, these architectures consist of a collection of latent states that

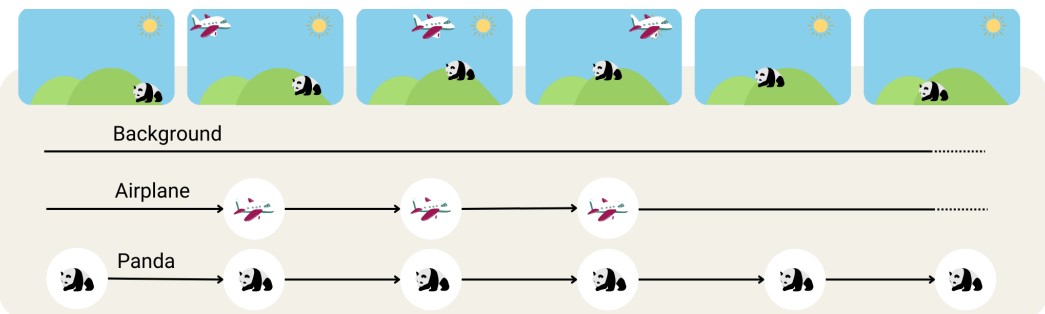

Figure 1: Videos can be viewed as a collection of features organised in a spatiotemporal hierarchy. This graphic illustrates a sequence of frames, in which the components of the video possess different temporal dynamics (white circles indicate feature changes). Notice the irregularities in their dynamics – the panda continuously changes its position, the airplane is seen only for a few timesteps, while the background remains static throughout. Similar to this, our model learns hierarchically disentangled representations of video features with the ability to model their unique dynamics.

transition over different timescales, which has been shown to benefit long-term predictions (Saxena et al., 2021; Zakharov et al., 2022).

Building upon these notions, we propose an architecture of a hierarchical generative model for long-horizon video prediction – *Dynamic Latent Hierarchy* (DLH). The principle ideas underlying this work are two-fold. First, we posit that learning disentangled hierarchical representations and their separate temporal dynamics increases the model's expressivity and breaks down the problem of video modelling into simpler sub-problems, thus benefiting prediction quality. As such, our model is capable of discovering the appropriate hierarchical spatiotemporal structure of the dataset, seamlessly adapting its generative structure to a dataset's dynamics. Second, the existence of a spatiotemporal hierarchy, in which some features can remain static for an arbitrary period of time (e.g. background in Fig. 1), implies that predicting the next state at every timestep may introduce unnecessary accumulation of error and computational complexity. Instead, our model learns to transition between states only if a change in the represented features has been observed (e.g. airplane in Fig. 1). Conversely, if no change in the features has been detected, the model clusters such temporally-persistent states closer together, thus building a more organised latent space. Our contributions are summarised as follows:

- A novel architecture of a hierarchical latent-variable generative model employing temporal Gaussian mixtures (GM) for representing latent states and their dynamics;
- Incorporation of a non-parametric inference method for estimating the discrete posterior distribution over the temporal GM components;
- The resulting superior long-horizon video prediction performance, emergent hierarchical disentanglement properties, and improved stochasticity representation.

## 2 DYNAMIC LATENT HIERARCHY

We propose an architecture of a hierarchical latent model for video prediction – *Dynamic Latent Hierarchy*. DLH consists of a hierarchy of latent states that evolve over different and flexible timescales. Each latent state is a mixture of two Gaussian components that represent the *immediate past* and the *predicted future* in a single belief state. Using this formalisation, the model dynamically assigns every newly inferred posterior state to one of these clusters, and thus implicitly learns the temporal hierarchy of the data in an unsupervised fashion.

### 2.1 GENERATIVE MODEL

We consider sequences of observations, $\{\mathbf{o}_1, ..., \mathbf{o}_T\}$, modelled by a hierarchical generative model with a joint distribution in the form (Fig. 2),

$$\prod_{t=1}^{T} p_\theta(\mathbf{o}_t, \vec{\mathbf{s}}_t, \vec{\mathbf{e}}_t) = \prod_{t=1}^{T} p_\theta(\mathbf{o}_t | \vec{\mathbf{s}}_t) \prod_{n=1}^{N} p_\theta(\mathbf{s}_t^n | \underbrace{\mathbf{e}_t^n}_{\text{indicator}}, \underbrace{\mathbf{s}_{<t}^n}_{\text{temporal}}, \underbrace{\mathbf{s}_t^{>n}}_{\text{hierarchical}}) p_\theta(\mathbf{e}_t^n | \mathbf{s}_{<t}^n), \qquad (1)$$

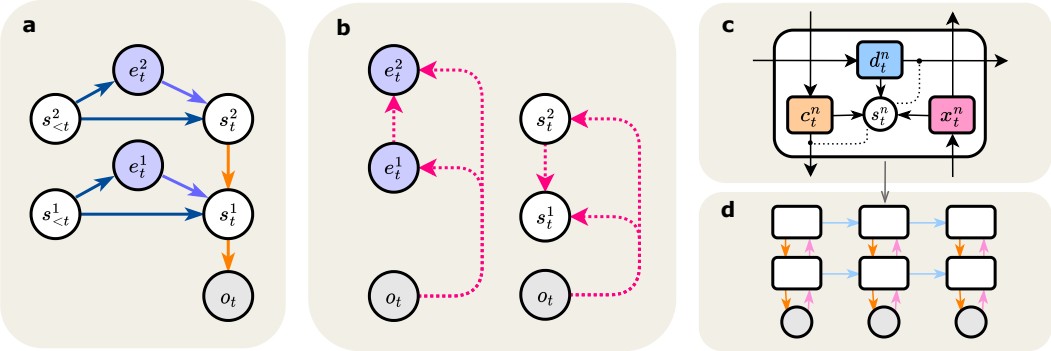

Figure 2: **(a)** Generative model of DLH. **(b)** Inference models of DLH. **(c)** Architectural components of DLH, showing the deterministic variables that correspond to bottom-up, top-down, and temporal information. **(d)** Example of a two-level DLH model rolled-out over three timesteps.

where $\mathbf{s}_t^n \sim \mathcal{N}(\cdot, \cdot)$ is a diagonal Gaussian latent state, $\mathbf{e}_t^n \sim \text{Ber}(\cdot)$ is the corresponding Bernoulli variable at a hierarchical level $n$ and timestep $t$, $\vec{\mathbf{s}} = \{\mathbf{s}^1, ..., \mathbf{s}^N\}$ and $\vec{\mathbf{e}} = \{\mathbf{e}^1, ..., \mathbf{e}^N\}$ denote collections of all variables in a hierarchy, and $\theta$ are the parameters of the model. Notice that each state $\mathbf{s}_t^n$ is conditioned on all of the hierarchical states above, past states in the same level, and an indicator variable $\mathbf{e}_t^n$.

One of the key features of DLH is the representation of a latent state as a temporal mixture of Gaussians (MoG). In particular, variables $\mathbf{s}_t^n$ and $\mathbf{e}_t^n$ together define a MoG, $p(\mathbf{s}_t^n, \mathbf{e}_t^n) = p(\mathbf{s}_t^n|\mathbf{e}_t^n)p(\mathbf{e}_t^n)^1$, with just two components such that,

$$p(\mathbf{s}_t^n|\mathbf{e}_t^n) = \begin{cases} p(\mathbf{s}_{t-1}^n) & \text{if } \mathbf{e}_t^n = 0, \text{ (previous state: } \textit{static prior}\text{)} \\ p_\theta(\mathbf{s}_t^n|\mathbf{s}_{<t}^n) & \text{if } \mathbf{e}_t^n = 1, \text{ (predicted state: } \textit{change prior}\text{)}. \end{cases} \tag{2}$$

As such, at every timestep, DLH holds two prior beliefs over the state of the world: (1) it can remain *static*, or (2) it can progress through time and thus *change*. In this view, variable $\mathbf{e}_t^n$ can be informally described as the probability of whether state $\mathbf{s}_t^n$ should be updated or remain fixed at timestep $t$ (Fig. 3). This property allows DLH to model data as a collection of hierarchical latent states that evolve over different and flexible timescales, determined by the indicator variable $\mathbf{e}^n$.

## 2.2 INFERENCE

In order to train the model using a variational lower bound, we must estimate the posterior distribution $q(\mathbf{s}_t^n, \mathbf{e}_t^n|\mathbf{o}_t)$, for which we assume a mean-field factorisation $q(\mathbf{s}_t^n)q(\mathbf{e}_t^n)$; therefore, the two distributions are approximated separately (Fig. 2b).

**Estimating** $q(\mathbf{s})$    In DLH, posterior $q(\mathbf{s}_t^n)$ is assumed to be a diagonal Gaussian, amortised using a neural network $q_\psi(\mathbf{s}_t^n|\mathbf{s}_t^{>n}, \mathbf{o}_t)$ with parameters $\psi$, conditioned on hierarchical states above and the latest data point $\mathbf{o}_t$. In line with the established procedure, the approximate posterior is trained using the reparametrisation trick (Kingma & Welling, 2014).

**Estimating** $q(\mathbf{e})$    Using reparametrisation tricks for discrete latent variables poses a significant challenge for a stable training procedure of deep learning models, which can be further exacerbated in hierarchical models (Falck et al., 2021). To avoid this, we estimate $q(\mathbf{e})$ using a non-parametric method.

Inferring distribution $q(\mathbf{e}^n)$ can be conceptualised as a clustering problem of $q(\mathbf{s}^n)$ with respect to the *static* and *change* priors of the model, with the central question being: under which temporal mixture component in Eq. 2 is the inferred state most likely? Has the state of the world changed or has it remained the same? As such, we formulate the approximation of $p(\mathbf{e}^n|\mathbf{s}^n)$ as model selection using expected likelihood ratio, where the two components of the MoG (Eq. 2) are the competing

---

[1]Stripping away the hierarchical and temporal conditioning for clarity.

models. Under the inferred state, $q'(\mathbf{s}_t^n) = q_\psi(\mathbf{s}_t^n|\mathbf{s}_t^{>n}, \mathbf{o}_t)$, the expected log-likelihood ratio is,

$$\mathbb{E}[\log \Lambda(\mathbf{s}_t^n)] = \mathbb{E}_{q'(\mathbf{s}_t^n)} \log \frac{p'(\mathbf{s}_t^n|\mathbf{e}_t^n = 0)}{p'(\mathbf{s}_t^n|\mathbf{e}_t^n = 1)}, \tag{3}$$

where $p'(\mathbf{s}_t^n|\mathbf{e}_t^n) = p_\theta(\mathbf{s}_t^n|\mathbf{e}_t^n, \mathbf{s}_{<t}^n, \mathbf{s}_t^{>n})$ from the definition of the generative model. Assuming the selection of the most likely component under the inferred posterior, we come to the following selection criterion (see full derivation and further clarifications in Appendix B),

$$D_{\mathrm{KL}}[q'(\mathbf{s}_t^n)||p'(\mathbf{s}_t^n|\mathbf{e}_t^n = 1)] \underset{\mathbf{e}_t^n = 0}{\overset{\mathbf{e}_t^n = 1}{\lessgtr}} D_{\mathrm{KL}}[q'(\mathbf{s}_t^n)||p'(\mathbf{s}_t^n|\mathbf{e}_t^n = 0)], \tag{4}$$

where the most likely component $i$ is approximated to have a probability $q(\mathbf{e}_t^n = i) = 1$. This approximation relates to the VaDE trick, which is similarly a non-parametric method of estimating the posterior component variable of a MoG (Jiang et al., 2017; Falck et al., 2021). In particular, our method can be viewed as taking a sample from the most likely component of the VaDE-estimated $q(\mathbf{e}^n)$ under the assumption of equal prior probabilities (see Appendix B.3). Though this method introduces bias, in practice, we found that it performs better than the VaDE trick. We hypothesise that this relates to a relatively fast convergence of the parametrised prior $p_\theta(\mathbf{e}_t^n|\cdot)$ model, which becomes overly confident in its predictions (even before any video features have been learned), thus irreversibly skewing the approximation of $q(\mathbf{e}_t^n)$. Nevertheless, we believe this direction of future work may merit further investigation.

### 2.3 NESTED TIMESCALES

We add a constraint on the hierarchical temporal structure of the generative model similar to Saxena et al. (2021); Zakharov et al. (2022). In particular, $q(\mathbf{e}^{n+1} = 1|\mathbf{e}^n = 0) = 0$. Enforcing this constraint has been shown to be an effective method to promote spatiotemporal disentanglement of features in hierarchical models, encouraging the representation of progressively slower features in the higher levels of the model. Furthermore, to reduce the computational complexity of the model, we block any further inference above the hierarchical level where $\mathbf{e}^n = 0$ is inferred, such that:

$$\text{if } \mathbf{e}_t^{n-1} = 0, \text{ then } \mathbf{e}_t^n = 0 \text{ and } q(\mathbf{s}_t^n) \leftarrow q(\mathbf{s}_{t-1}^n). \tag{5}$$

Lastly, to model continuously changing videos, we assume $q(\mathbf{e}^1 = 1) = 1$, which allows for the bottom level of DLH to always be in use. It is worth noting that the proposed model constraints may be relaxed in different implementations of DLH, which could be explored in future work.

### 2.4 LOWER BOUND ESTIMATION

To train the model, we derive a variational lower bound (ELBO), for which we introduce an approximate posterior distribution $q(\vec{\mathbf{s}}, \vec{\mathbf{e}})$ so that

$$\sum_{t=1}^T \log p(\mathbf{o}_t) = \sum_{t=1}^T \log \int_{\vec{\mathbf{s}}} \sum_{\vec{\mathbf{e}}} q(\vec{\mathbf{s}}_t, \vec{\mathbf{e}}_t) \frac{p_\theta(\mathbf{o}_t, \vec{\mathbf{s}}_t, \vec{\mathbf{e}}_t)}{q(\vec{\mathbf{s}}_t, \vec{\mathbf{e}}_t)} \tag{6}$$

$$\geq \sum_{t=1}^T \mathbb{E}_{q(\vec{\mathbf{s}}_t, \vec{\mathbf{e}}_t)} \log p_\theta(\mathbf{o}_t|\vec{\mathbf{s}}_t) + \mathbb{E}_{q(\vec{\mathbf{s}}_t, \vec{\mathbf{e}}_t)} \Big[ \log \frac{p_\theta(\vec{\mathbf{s}}_t|\vec{\mathbf{e}}_t) p_\theta(\vec{\mathbf{e}}_t)}{q(\vec{\mathbf{s}}_t, \vec{\mathbf{e}}_t)} \Big]. \tag{7}$$

Assuming posterior factorisation of $q(\mathbf{s}_t^n, \mathbf{e}_t^n) = q_\psi(\mathbf{s}_t^n|\mathbf{s}_t^{>n}, \mathbf{o}_t) q(\mathbf{e}_t^n|\mathbf{e}_t^{n-1})$, we write the complete formulation of the ELBO,

$$\mathcal{L}_{\mathrm{ELBO}} = \sum_{t=1}^T \Big[ \mathbb{E}_{q_\psi(\vec{\mathbf{s}}_t)} \log p_\theta(\mathbf{o}_t|\vec{\mathbf{s}}_t) \Big] \tag{8a}$$

$$- \sum_{t=1}^T \sum_{n=1}^N \Big[ \mathbb{E}_{q(\mathbf{e}_t^n)q_\psi(\mathbf{s}_{<t}^n, \mathbf{s}_t^{>n})} D_{\mathrm{KL}} \big[ q_\psi(\mathbf{s}_t^n|\mathbf{s}_t^{>n}, \mathbf{o}_t)||p_\theta(\mathbf{s}_t^n|\mathbf{e}_t^n, \mathbf{s}_{<t}^n, \mathbf{s}_t^{>n}) \big] \Big] \tag{8b}$$

$$- \sum_{t=1}^T \sum_{n=1}^N \Big[ \mathbb{E}_{q(\mathbf{e}_t^{n-1})q_\psi(\mathbf{s}_{<t}^n)} D_{\mathrm{KL}} \big[ q(\mathbf{e}_t^n|\mathbf{e}_t^{n-1})||p_\theta(\mathbf{e}_t^n|\mathbf{s}_{<t}^n) \big] \Big]. \tag{8c}$$

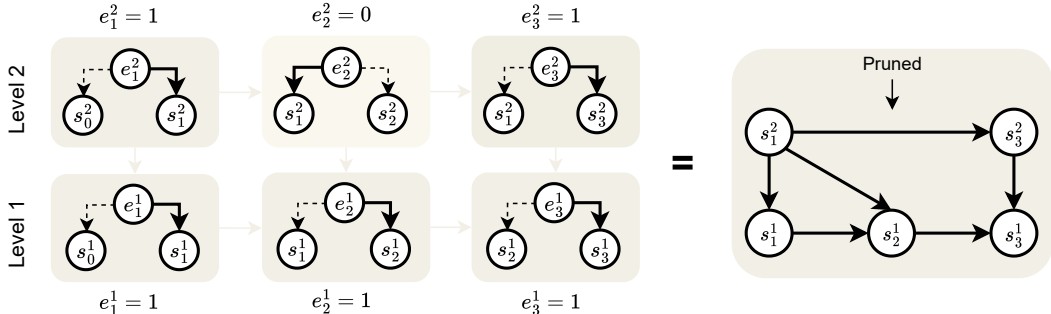

Figure 3: Sampling MoG components can be seen as changing the generative structure of the model. The diagram shows a two-level DLH unrolled over three timesteps – each box indicating a temporal MoG. Bold arrows indicate the sampled component of a MoG. At $n, t = 2$, the model samples the static component $0 \sim p(\mathbf{e}_2^2)$, thus state $\mathbf{s}_1^2$ remains fixed before being updated to $\mathbf{s}_3^2$ (indicated by the 'Pruned' label on the right).

To better understand the optimisation objective and the role of a temporal Gaussian mixture from Eq. 2, it is useful to break the down the three components of the ELBO. First, component 8a is the likelihood of the data under the inferred posteriors $q(\vec{\mathbf{s}}_t)$, which improves the quality of frame reconstructions. Second, component 8c is the KL divergence between the posterior and prior Bernoulli distributions over $\mathbf{e}$, allowing the parametrised prior model to learn the evolution of static and change priors over time. Lastly, component 8b regularises the latent belief space by bringing the posterior either closer to the static or to the change component of a prior MoG. This can be seen more clearly if we expand the expectation,

$$= q(\mathbf{e}_t^n = 0) \underbrace{D_{\mathrm{KL}}\big[q'(\mathbf{s}_t^n)||p'(\mathbf{s}_t^n|\mathbf{e}_t^n = 0)\big]}_{\text{posterior} \leftrightarrow \text{static prior}} + q(\mathbf{e}_t^n = 1) \underbrace{D_{\mathrm{KL}}\big[q'(\mathbf{s}_t^n)||p'(\mathbf{s}_t^n|\mathbf{e}_t^n = 1)\big]}_{\text{posterior} \leftrightarrow \text{change prior}}. \quad (9)$$

Depending on the inferred posterior distribution $q(\mathbf{e}_t^n)$, the model will employ the appropriate part of Eq. 9 in the optimisation. For example, if inferred that state $\mathbf{s}_t^n$ has not changed ($\mathbf{e}_t^n = 0$), the model will bring the new posterior and the static prior closer together, and vice versa. Ultimately, this allows the model to naturally cluster similar temporal states together, while learning to transition between states that are sufficiently separated in the belief space.

## 2.5 MODEL COMPONENTS

DLH's architectural implementation is similar to that of NVAE (Vahdat & Kautz, 2020) and VPR (Zakharov et al., 2022), which employ deterministic variables for propagating information through the model. More specifically, DLH consists of the following model components,

| | | | | | |
|---|---|---|---|---|---|
| Encoder, | $x_t^{n+1} = f_{\mathrm{enc}}^n(x_t^n)$ | (10) | Posterior state, | $q_\psi(\mathbf{s}_t^n|x_\tau^n, c_t^n)$ | (13) |
| Decoder, | $c_t^{n-1} = f_{\mathrm{dec}}^n(\mathbf{s}_t^n, c_t^n)$ | (11) | Prior state, | $p_\theta(\mathbf{s}_t^n|d_t^m, c_t^n)$ | (14) |
| Temporal, | $d_{t+1}^n = f_{\mathrm{tem}}^n(\mathbf{s}_t^n, d_t^n)$ | (12) | Prior factor, | $p_\theta(\mathbf{e}_t^n|d_t^m)$ | (15) |

where deterministic variables $x_t^n$, $c_t^n$, $d_t^n$ correspond to the bottom-up, top-down, and temporal variable transformations, as shown in Figure 2c, and $x_t^0$ and $c_t^0$ correspond to the output image $\mathbf{o}_t$. Variables $c_t^n$ and $d_t^n$ are non-linear transformations of samples from $\mathbf{s}_t^{>n}$ and $\mathbf{s}_{<t}^n$, respectively. We use a GRU model (Cho et al., 2014) for the transition and prior factor models, convolutional layers for the encoder and decoder, and fully-connected MLP layers for all other models.

## 3 RELATED WORK

**Video prediction** Early works in video prediction largely focused on different variants of deterministic models (Oh et al., 2015; Finn et al., 2016; Byravan & Fox, 2017; Vondrick & Torralba, 2017); however, it has been widely suggested that these models are poorly suited for capturing stochasticity that is often present in video datasets.

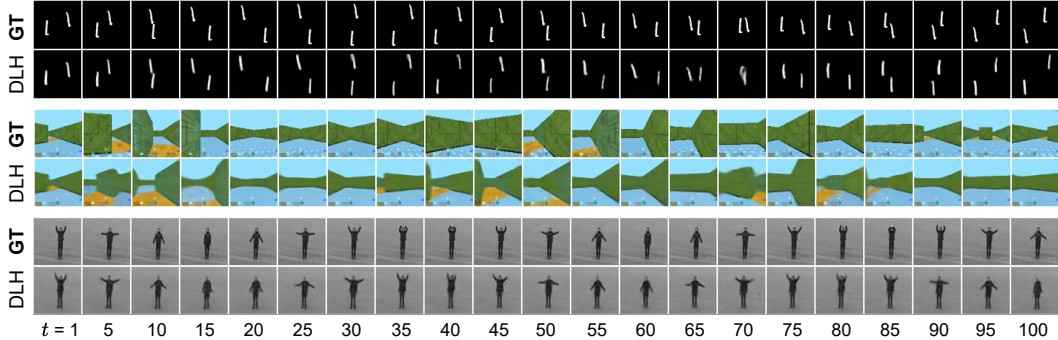

Figure 4: Open-loop video prediction given 30 context frames (not shown). GT: ground-truth sequence, DLH: our model's prediction. DLH maintains the important contextual information about the video, without significant degeneration in the reconstruction quality.

The problem of stochastic video prediction has been addressed using a variety of generative architectures. Models autoregressive in image space (Babaeizadeh et al., 2021; Reed et al., 2017; Weissenborn et al., 2020; Kalchbrenner et al., 2016; Denton & Fergus, 2018) demonstrate good results but suffer from high computational complexity, particularly for long-term predictions. GAN-based (Goodfellow et al., 2014) approaches have been popular due to their ability to produce sharp predictions (Clark et al., 2019; Arr; Mathieu et al., 2016; Lee et al., 2018). More recently, transformers (Vaswani et al., 2017) have been used to model video datasets, both in latent (Rakhimov et al., 2020; Yan et al., 2021) and pixel space (Weissenborn et al., 2020). A fairly large category of video architectures is based on using Variational Autoencoders (Kingma & Welling, 2014), which have been shown to produce meaningful latent representations on image (Vahdat & Kautz, 2020; Higgins et al., 2022) and video data (Zakharov et al., 2022). Variational autoencoders (VAE)-based models that attempt to learn temporal dependencies in the latent space (Wu et al., 2021; Villegas et al., 2019; Castrejon et al., 2019; Franceschi et al., 2020; Saxena et al., 2021; Yan et al., 2021; Zakharov et al., 2022) generate good performance but generally suffer from blurry predictions, referred to as the 'underfitting problem' (Babaeizadeh et al., 2021; Wu et al., 2021; Villegas et al., 2019). Nevertheless, these models benefit from computational efficiency since the learning of temporal video dynamics commonly happens in a lower-dimensional latent space. Most recently, diffusion models have been shown to produce great performance on both short (Yang et al., 2022; Höppe et al., 2022) and long (Harvey et al., 2022) videos.

**Hierarchical generative models**   Hierarchical generative models have been shown to be an effective way of modelling high-dimensional data, including images (Rasmus et al., 2015; Sønderby et al., 2016; Maaløe et al., 2019; Vahdat & Kautz, 2020; Child, 2021) and videos (Saxena et al., 2021; Kim et al., 2019; Pertsch et al., 2020; Hsu et al., 2019; Zakharov et al., 2022), producing rich latent representations and demonstrating strong representational power.

**Temporal abstraction**   The topic of learning temporal abstractions from sequential data has been harmoniously rising in popularity alongside the progress in deep and hierarchical latent models. Temporal abstraction models often operate a number of hierarchical latent variables updating over different timescales, with the goal of capturing the temporal features of a dataset (Chung et al., 2017; Mujika et al., 2017; Kim et al., 2019; Saxena et al., 2021; Fountas et al., 2022; Zakharov et al., 2022), though other models learn the relevant prediction timescales without resorting to hierarchical methods (Chung et al., 2017; Neitz et al., 2018; Jayaraman et al., 2018; Shang et al., 2019; Kipf et al., 2019; Pertsch et al., 2020; Zakharov et al., 2021).

**Gaussian Mixtures in VAEs**   Our work similarly touches on the topic of VAEs with Gaussian Mixture latent states. Generally, these models are aimed at producing meaningful structure of the latent space, in which data points are clustered in an unsupervised fashion (Dilokthanakul et al., 2016; Jiang et al., 2017; Falck et al., 2021). Though highly relevant conceptually, these works deal with non-temporal data and therefore have fundamentally different formulations.

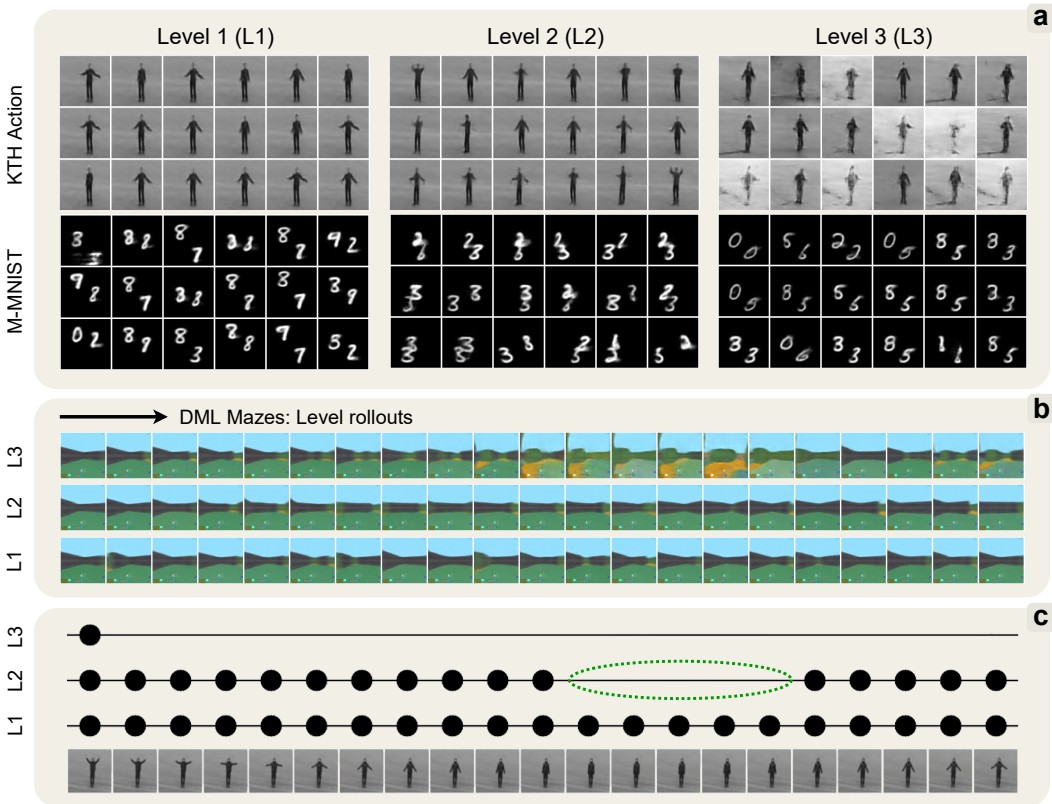

Figure 5: Hierarchical disentanglement in DLH. **(A)**: Random samples independently drawn from the different hierarchical levels while keeping all other levels fixed. They qualitatively illustrate what information is contained within a particular level. In KTH, L1 and L2 tend to encode motion; L3 encodes the general context of the frame. In M-MNIST, L1 encodes slight variations in the position and digits; L2 represents position; L3 contains both style and digit types. **(B)**: Rolling out hierarchical levels in the DML Mazes under all other levels being fixed. L1 includes minor variations in the view angle; L2 changes the position of the observer and wall shape; L3 predicts transitions to the different parts of the maze. **(C)**: Inferred components of the temporal MoGs at every level of the model. Black circles indicate $\mathbf{e}^n = 1$ (change component) inferred by DLH. L2 infers the static component when the person's arms are not moving (green ellipse). Similarly, L3 is static throughout the duration of the video.

## 4 EXPERIMENTS

In this section, we showcase the representational properties of DLH, and their resulting impact on the performance of the model for long-term video prediction. In particular, we demonstrate that DLH: (a) outperforms benchmarks in long-term video prediction, (b) produces an organised hierarchical latent space with spatiotemporal disentanglement and temporal abstraction, (c) generates coherent videos even in datasets characterised by temporal stochasticity, and (d) dynamically regulates its structural complexity. In the analysis probing DLH's expressivity and representations, we emphasise how the presented formulation of the generative model, in particular the use of temporal MoG, naturally results in the emergent properties of the model.

### 4.1 DATASETS AND BENCHMARKS

**Datasets** To test the model's ability in long-term video prediction, we use Moving MNIST (Srivastava et al., 2016) with 300 timesteps, KTH Action (Schuldt et al., 2004) with 300 timesteps, and DeepMind Lab (DML) Mazes (Eslami et al., 2018) with 200 timesteps. For a more detailed analysis of the model's properties, we use a toy Moving Ball dataset (Zakharov et al., 2022).

**Benchmarks** Clockwork Variational Autoencoder (CW-VAE) (Saxena et al., 2021) is a hierarchical VAE for video prediction, in which latent variables operate over fixed-temporal schedules, similarly subject to nested timescales. CW-VAE demonstrated state-of-the-art performance in long-term video

prediction, indicating the merit of the slower-evolving context states. VTA (Kim et al., 2019) is a two-level hierarchical model for video prediction that employs a parametrised boundary detector to learn sub-sequences of a video and generate temporally-abstracted representations. LMC-Memory (Lee et al., 2021) learns and stores long-term motion context for better long-horizon video prediction, which has been shown to outperform other RNN-based approaches.

**Metrics**  To evaluate stochastic video prediction, we employ the standard procedure of sampling 100 conditionally generated sequences and picking the best one to report (Denton & Fergus, 2018). For metrics, we use Structural Similarity (SSIM) and Peak Signal-to-Noise Ratio (PSNR) to test the performance of a model with respect to the ground-truth videos.

Table 1: Open-loop video prediction. Stars denote <5% standard deviation.

| **M-MNIST** | SSIM↑ | PSNR↑ |
|---|---|---|
| DLH (Ours) | **0.78**\* | **15.7**\* |
| CW-VAE | 0.68\* | 13.11\* |
| VTA | 0.58 | 12.18 |
| LMC-Memory | 0.75\* | 13.73\* |
| **KTH Action** | SSIM↑ | PSNR↑ |
| DLH (Ours) | **0.84**\* | **24.7** |
| CW-VAE | 0.80 | 22.0 |
| VTA | 0.77 | 22.41 |
| LMC-Memory | 0.83\* | 23.44 |
| **DML Mazes** | SSIM↑ | PSNR↑ |
| DLH (Ours) | **0.59** | **14.3** |
| CW-VAE | 0.44 | 13.71\* |
| VTA | 0.55 | 13.51\* |

Parameter count: **DLH** (7M), CW-VAE (12M), VTA (3M), LMC-Memory (34M).

## 4.2 Video prediction and generation

Table 1 shows the evaluation of DLH and its benchmarks in the task of long-horizon video prediction. As evident, DLH outperforms other models across the presented datasets. Figure 4 shows some examples of long-horizon open-loop rollouts. For Moving MNIST, DLH maintains the information about the digits throughout the sequence, while also accurately predicting their positions. For DML Mazes, DLH correctly predicts the colours and wall positions, without switching to a configuration of another maze. Similarly, for KTH, our model preserves the important contextual knowledge (e.g. background) and accurately predicts the long sequence of arm swings. Appendices C.1 and C.2 include qualitative comparisons of the models and the per-frame metric plots.

## 4.3 Hierarchical abstraction and dynamic structure

DLH exhibits characteristics of a model that learns temporally-abstracted and hierarchically disentangled representations. Figure 5a demonstrates reconstructed frames retrieved by sampling the different hierarchical levels of the model. Here, we observe the variations in the samples that correspond to meaningful and interpretable spatiotemporal features of the videos. In Figure 5b, we show rollouts of the model's levels (other levels being fixed) using DML Mazes, which indicate that DLH learns to transition between progressively slower features in the higher levels of its hierarchy.

Figure 5c demonstrates another telling qualitative evaluation of DLH and its representations – the inferred components of $\mathbf{e}^n$ (static or change) for a given video. In particular, it shows that the model continuously detects feature changes in the second level of its hierarchy (L2) when the person is moving their arms, and conversely when the person's are motionless. Furthermore, it can be seen that the the top level (L3) remains static throughout. Notably, these results are in agreement with the random samples shown above, and more clearly illustrate the property of hierarchical disentanglement present in the model.

Table 2: Average number of employed levels ($\bar{L}$) and the total KL loss in the instances of DLH with different number of hierarchical levels (Moving Ball)

| Levels | $\bar{L}$ | KL loss |
|---|---|---|
| 2 | $1.22 \pm 0.05$ | $41.9 \pm 0.3$ |
| 3 | $1.24 \pm 0.05$ | $42.8 \pm 1.1$ |
| 4 | $1.32 \pm 0.12$ | $41.1 \pm 1.0$ |
| 5 | $1.38 \pm 0.07$ | $43.1 \pm 2.1$ |

The capacity of DLH to learn the spatiotemporal representation of features along its hierarchy is largely driven by the dynamic manipulation of its hierarchical and temporal structure (Figure 3). Interestingly, we observe that DLH consistently converges to similar structures even when possessing different number of levels. Table 2 shows the average hierarchical depth employed by the model ($\bar{L} = \frac{1}{T} \sum_{t=1}^{T} \sum_{n=1}^{N} \mathbf{e}_t^n$) over a video length given the total number of hierarchical levels it has (trained using the Moving Ball dataset). As evident, the models converge to similar values despite their size differences, indicating that DLH naturally simplifies its structure and does not employ

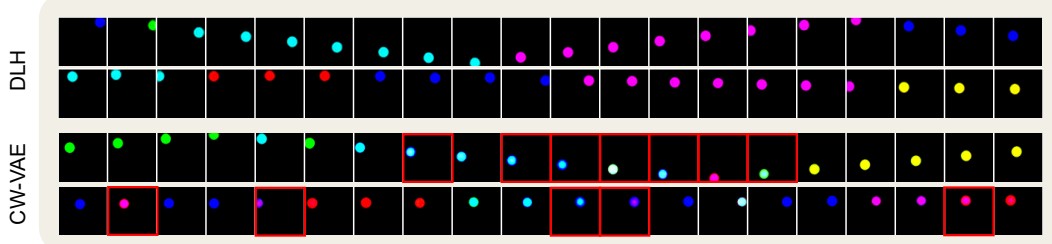

Figure 6: Two randomly-selected open-loop rollouts in a stochastic Moving Ball dataset using DLH (top) and CW-VAE (bottom) models. DLH successfully produces realistic rollouts, while CW-VAE struggles to produce rollouts with sharp and random colour changes (frames highlighted in red).

more resources than necessary. This is similarly substantiated by the comparable magnitudes of the total KL loss, which is commonly used to indicate the amount of information stored in the latent states (see Appendix C.3 for a more in-depth analysis).

## 4.4 TEMPORAL STOCHASTICITY

Videos often contain temporal stochasticity, where features may change at seemingly random times. How would a generative model represent such uncertainty? In the context of employing a Gaussian latent state, the uncertainty about the next state would have to be reflected in the higher variance, in order to cover both possible outcomes; however, this necessarily increases the chance of sampling areas of the latent space that do not correspond to any meaningful states, harming the prediction performance. In DLH, by virtue of the temporal MoG, such stochasticity can be effectively captured by variable $\mathbf{e}^n$, which decides whether the latent state $\mathbf{s}^n$ should be updated or remain fixed, alleviating the need to sample from degener-

Table 3: Average predicted probability of $p_\theta(\mathbf{e}^2 = 1)$ under the different levels of temporal stochasticity ($\lambda$) in the Moving Ball dataset

| $\lambda$ | $p_\theta(\mathbf{e}^2 = 1)$ | |
|---|---|---|
| | change | static |
| 0.0 | $.97 \pm 0.01$ | $.007 \pm .003$ |
| 0.1 | $.82 \pm 0.02$ | $.074 \pm .007$ |
| 0.3 | $.73 \pm 0.02$ | $.167 \pm .011$ |

ate regions of the latent space. To demonstrate this, we modify the Moving Ball dataset to include random colour changes that can occur at every timestep with a probability of $\lambda$ and train a two-level DLH using it. Figure 6 shows a comparison of open-loop rollouts generated by DLH and CW-VAE, trained on the stochastic Moving Ball with $\lambda = 0.1$. While CW-VAE struggles to generate rollouts with consistent and sharp colour changes, DLH faces no such problems, producing sequences with both deterministic and random colour switches. This experiment shows the important role of MoGs in representing temporal stochasticity, and highlights the superior representational capacity of DLH.

Similarly, the behaviour of prior $p_\theta(\mathbf{e}^2|\cdot)$ under temporal stochasticity can be more clearly understood using the results in Table 3, which shows the average predicted probability of the change component, at level 2, under the inferred posterior $q(\mathbf{e}^2)$ being either *change* ($\mathbf{e}^2 = 1$) or *static* ($\mathbf{e}^2 = 0$). More stochasticity necessarily implies the reduced ability to predict when the observed video features would change (as would be signalled by the inferred posterior component), which should be reflected in the average probabilities predicted by the prior component model. Indeed, in Table 3, we observe that as the stochasticity of the dataset, $\lambda$, rises, the model becomes more cautious in its predictions.

## 5 DISCUSSION

Our work demonstrates that building generative models with better representational properties, such as spatiotemporal and hierarchical disentanglement, translates to better predictive capabilities in long and complex time series. Furthermore, we believe that improving the quality of latent representations is of high importance for model-based reinforcement learning agents, where accurate predictions of the future lead to better planning and offline credit assignment, while a hierarchical and nested treatment of time could allow for temporally-abstract reasoning. Nevertheless, one of the limitations facing VAE-based models, and by extension our own, is the lack of sharpness in the predictions. Though significant progress has been made in the recent years (Babaeizadeh et al., 2021; Wu et al., 2021), addressing this problem in DLH can be a significant next step for further improving the performance of the model.

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

## A   TRAINING AND ARCHITECTURAL DETAILS

### A.1   IMPLEMENTATION AND TRAINING

For all datasets, DLH was trained with video sequences of length 100 and batch 100. We used Adam optimiser (Kingma & Ba, 2015) with a learning rate of $0.0005$ and $\epsilon = 0.0001$. We also find that it is beneficial to multiply the KL loss with a parameter $\beta$, which is slowly increased to the value of $1.0$ over the course of the first 10k training iterations. This promotes the model to learn good reconstructions before being severely restricted in increasing its latent complexity. The image reconstruction model predicts the means of a Gaussian with fixed standard deviation, which is optimised separately for each dataset.

For this paper, we used a three-level DLH for all datasets (except for Moving Ball, where we used just two levels). Each level possesses deterministic variables, for which we set $|x^n| = |d^n| = |c^n| = 200$, and random variables, where $|\mathbf{s}^n| = 20$. For encoding and decoding image data, we use 4 convolutional and 4 transposed-convolutional layers, respectively. Table 4 shows the architectural details of all other model components, including the sizes of neural networks. The total number of parameters of a three-level DLH is 7M, and its training takes $\sim$3 days on a single Tesla V100.

Table 4: Architecture of models from Eq.10-15.

| Model | Hidden layers |
|---|---|
| Encoder, $\mathrm{f}_{\mathrm{enc}}^n(x_t^n)$ | MLP [200] |
| Decoder, $\mathrm{f}_{\mathrm{dec}}^n(\mathbf{s}_t^n, c_t^n)$ | MLP [200] |
| Temporal, $\mathrm{f}_{\mathrm{tem}}^n(\mathbf{s}_t^n, d_t^n)$ | GRU [200] |
| Posterior, $\mathrm{q}_\psi(\mathbf{s}_t^n|x_\tau^n, \mathbf{s}_t^{>n})$ | MLP [40, 40, 40] |
| Prior state, $\mathrm{p}_\theta(\mathbf{s}_t^n|\mathbf{s}_{<t}^n, \mathbf{s}_t^{>n})$ | MLP [40, 40, 40] |
| Prior factor, $\mathrm{p}_\theta(\mathbf{e}_t^n|\mathbf{s}_{<t}^n)$ | GRU [200] |

### A.2   PSEUDOCODE

---

**Algorithm 1:** Pseudocode of DLH

---

**for** $\{\boldsymbol{o}_{1:T}\}_i \sim \mathcal{D}$ **do**

    **for** $t = 1$ **to** $T$ **do**

        **for** $n = 1$ **to** $N$ **do**

            $\triangleright$ Compute the log-likelihood ratio under approximate posterior $\mathbb{E}_{p'(\mathbf{s}_t^{>n}|\mathbf{e}_t^{>n}=0)} q_\psi(\mathbf{s}_t^n|\mathbf{s}_t^{>n}, \mathbf{o}_t)$ using $D_{\mathrm{KL}}[q_\psi(\mathbf{s}_t^n)||p'(\mathbf{s}_t^n|\mathbf{e}_t^n = 1)]$ and $D_{\mathrm{KL}}[q_\psi(\mathbf{s}_t^n)||p'(\mathbf{s}_t^n|\mathbf{e}_t^n = 0)]$.

            $\triangleright$ Approximate the posterior $q(\mathbf{e}_t^n|\mathbf{e}_t^{n-1})$ using Eq. 4 and the nested timescales constraint, $q(\mathbf{e}_t^n = 1|\mathbf{e}_t^{n-1} = 0) = 0$.

            **if** $0 \sim q(\mathbf{e}_t^n)$ **then**

                $\triangleright$ Approximate $q(\mathbf{e}_t^{>n} = 1|\cdot) = 0$ and $q'(\mathbf{s}_t^{>n}) \leftarrow q'(\mathbf{s}_{t-1}^{>n})$

                $\triangleright$ $K \leftarrow n$

                **break**

        **for** $n = K$ **to** $1$ **do**

            $\triangleright$ Compute the state posterior $q_\psi(\mathbf{s}_t^n|\mathbf{s}_t^{>n}, \mathbf{o}_t)$

    $\triangleright$ Compute the ELBO (Eq. 8) using the inferred posteriors $q_\psi(\vec{\mathbf{s}}_t|\mathbf{o}_t)q(\vec{\mathbf{e}})$.

    $\triangleright$ Apply a gradient step on $\theta, \psi$.

---

# B APPROXIMATION OF THE POSTERIOR COMPONENT OF MOG

## B.1 EXPECTED LOG-LIKELIHOOD RATIO

In Section 2.2, we propose an approximation of the posterior $q(\mathbf{e}_t^n)$ using the expected log-likelihood ratio in Eq. 3. We can arrive at this formulation via either (a) considering the log-likelihood ratio of the two components of the MoG, or (b) the VaDE trick from the Gaussian Mixture VAE literature (Appendix B.3). In this section, we briefly explain the former perspective.

We start by considering the two components in the prior MoG as competing models. Their log-likelihood ratio is defined as,

$$\log \Lambda(\mathbf{s}^n) = \log \prod_{i=0}^{D} \frac{p'(\mathbf{s}_i^n|\mathbf{e}^n = 0)}{p'(\mathbf{s}_i^n|\mathbf{e}^n = 1)} = \sum_{i=0}^{D} \log \frac{p'(\mathbf{s}_i^n|\mathbf{e}^n = 0)}{p'(\mathbf{s}_i^n|\mathbf{e}^n = 1)}, \tag{16}$$

where $\mathbf{s}_i^n \sim q'(\mathbf{s}^n)$, $D$ is the number of posterior samples, and $\Lambda(\mathbf{s}^n) = \prod_{i=0}^{D} \frac{p'(\mathbf{s}_i^n|\mathbf{e}^n=0)}{p'(\mathbf{s}_i^n|\mathbf{e}^n=1)}$. By applying the Law of Large numbers, taking $D \to \infty$, we can write this as an expectation with respect to the posterior $q'(\mathbf{s}^n)$,

$$\mathbb{E}[\log \Lambda(\mathbf{s}^n)] = \mathbb{E}_{q'(\mathbf{s}^n)} \log \frac{p'(\mathbf{s}^n|\mathbf{e}^n = 0)}{p'(\mathbf{s}^n|\mathbf{e}^n = 1)}, \tag{17}$$

where the static and prior components, $p'(\mathbf{s}^n|\mathbf{e}^n = 0)$ and $p'(\mathbf{s}^n|\mathbf{e}^n = 1)$, are viewed as the competing models under the inferred posterior $q'(\mathbf{s}^n)$. This expected likelihood ratio can be computed in terms of KL divergences, if we add and subtract the entropy of $q'(\mathbf{s}^n)$,

$$\mathbb{E}[\log \Lambda(\mathbf{s}^n)] = \mathbb{E}_{q'(\mathbf{s}^n)} \log p'(\mathbf{s}^n|\mathbf{e}^n = 0) - \mathbb{E}_{q'(\mathbf{s}^n)} \log q'(\mathbf{s}^n) \tag{18}$$

$$- \mathbb{E}_{q'(\mathbf{s}^n)} \log p'(\mathbf{s}^n|\mathbf{e}^n = 1) + \mathbb{E}_{q'(\mathbf{s}^n)} \log q'(\mathbf{s}^n) \tag{19}$$

$$= D_{\mathrm{KL}}[q'(\mathbf{s}^n)||p'(\mathbf{s}^n|\mathbf{e}^n = 1)] - D_{\mathrm{KL}}[q'(\mathbf{s}^n)||p'(\mathbf{s}^n|\mathbf{e}^n = 0)]. \tag{20}$$

Assuming minimum probability of error test, the selection of the most likely component is realised via

$$D_{\mathrm{KL}}[q'(\mathbf{s}^n)||p'(\mathbf{s}^n|\mathbf{e}^n = 1)] - D_{\mathrm{KL}}[q'(\mathbf{s}^n)||p'(\mathbf{s}^n|\mathbf{e}^n = 0)] \underset{\mathbf{e}^n=0}{\overset{\mathbf{e}^n=1}{\lessgtr}} 0. \tag{21}$$

## B.2 EXPECTED LOG-LIKELIHOOD RATIO WITH HIERARCHICAL DEPENDENCIES

In DLH, posteriors are factorised hierarchically, such that

$$q(\vec{\mathbf{s}}_t) = \prod_{n=0}^{N} q_\psi(\mathbf{s}_t^n|\mathbf{s}_t^{>n}, \mathbf{o}_t). \tag{22}$$

In turn, this implies that the expected log-likelihood ratio must be computed with respect to the hierarchical posteriors,

$$\mathbb{E}[\log \Lambda(\mathbf{s}^n)] = \mathbb{E}_{q(\vec{\mathbf{s}}_t)} \log \frac{p'(\mathbf{s}^n|\mathbf{e}^n = 0)}{p'(\mathbf{s}^n|\mathbf{e}^n = 1)} \tag{23}$$

$$= \underbrace{\mathbb{E}_{q'(\mathbf{s}_t^{>n})}}_{\text{hierarchical context}} \mathbb{E}_{q'(\mathbf{s}_t^n)} \log \frac{p'(\mathbf{s}^n|\mathbf{e}^n = 0)}{p'(\mathbf{s}^n|\mathbf{e}^n = 1)}. \tag{24}$$

However, this estimation implies that *all* hierarchical posteriors, $q'(\mathbf{s}_t^{>n})$, must first be inferred in a top-down process, which may be computationally expensive, especially for large $N$. To resolve this, we approximate the hierarchical posteriors, above the level at which the posterior component $q(\mathbf{e}_t^n)$ is being estimated, using the static priors,

$$q'(\mathbf{s}_t^{>n}) \approx \prod_{j=n+1}^{N} p'(\mathbf{s}_t^j|\mathbf{e}_t^j = 0) = \prod_{j=n+1}^{N} q'(\mathbf{s}_{t-1}^j), \tag{25}$$

which are already known. This simple assumption alleviates the need to infer all hierarchical posteriors before the estimation of $q(\mathbf{e}_t^n)$ and has been shown to work well in practice.

More specifically, the computational savings come from the combination of two factors: (1) approximating hierarchical context using Eq. 25 during the inference of $q(\mathbf{e}_t^n)$, and (2) the nested timescales assumption in Eq. 5 that blocks any further inference beyond the level at which $\mathbf{e}_t^n = 0$. As such, the computations pertaining to the inference procedure are required only up to some level $k$, where $\mathbf{e}_t^k = 0$ is first inferred.

### B.3 RELATIONSHIP TO VADE TRICK

VaDE trick is a non-parametric technique for estimating posterior $p(\mathbf{e}|\mathbf{s})$ in Gaussian Mixture VAE models (Jiang et al., 2017; Falck et al., 2021). In particular, Falck et al. (2021) proved that an approximate posterior distribution $q(\mathbf{e})$ that minimises the KL divergence to the true posterior $p(\mathbf{e}|\mathbf{s})$ will take the form,

$$\arg\min_{q(\mathbf{e})} D_{\mathrm{KL}}[q(\mathbf{e})||p(\mathbf{e}|\mathbf{s})] = \frac{\exp(\mathbb{E}_{q(\mathbf{s})}\log p(\mathbf{e}|\mathbf{s}))}{\sum_{\mathbf{e}\in\mathbf{E}}\exp(\mathbb{E}_{q(\mathbf{s})}\log p(\mathbf{e}|\mathbf{s}))}, \tag{26}$$

where $K = |\mathbf{E}|$ is the number of components in a Gaussian mixture $p(\mathbf{s}, \mathbf{e})$, and $q(\mathbf{s})$ is an approximate posterior over $\mathbf{s}$.

In DLH, the number of components in a MoG is limited to $K = 2$. Using Eq. 26, we can compute the posterior odds and solve for one of the two components (e.g. $q(\mathbf{e} = 0)$),

$$\frac{q(\mathbf{e} = 0)}{q(\mathbf{e} = 1)} = \frac{\exp(\mathbb{E}_{q(\mathbf{s})}\log p(\mathbf{e} = 0|\mathbf{s}))}{\exp(\mathbb{E}_{q(\mathbf{s})}\log p(\mathbf{e} = 1|\mathbf{s}))} \tag{27}$$

$$= \exp\left[\mathbb{E}_{q(\mathbf{s})}\log p(\mathbf{e} = 0|\mathbf{s}) - \mathbb{E}_{q(\mathbf{s})}\log p(\mathbf{e} = 1|\mathbf{s})\right] \tag{28}$$

$$= \exp\left[\mathbb{E}_{q(\mathbf{s})}\log\frac{p(\mathbf{s}|\mathbf{e} = 0)p(\mathbf{e} = 0)}{\sum_{\mathbf{e}\in\mathbf{E}}p(\mathbf{s}|\mathbf{e})p(\mathbf{e})} - \mathbb{E}_{q(\mathbf{s})}\log\frac{p(\mathbf{s}|\mathbf{e} = 1)p(\mathbf{e} = 1)}{\sum_{\mathbf{e}\in\mathbf{E}}p(\mathbf{s}|\mathbf{e})p(\mathbf{e})}\right] \tag{29}$$

$$= \exp\left[\mathbb{E}_{q(\mathbf{s})}\log p(\mathbf{s}|\mathbf{e} = 0)p(\mathbf{e} = 0) - \mathbb{E}_{q(\mathbf{s})}\log p(\mathbf{s}|\mathbf{e} = 1)p(\mathbf{e} = 1)\right] \tag{30}$$

$$= \exp\left[\mathbb{E}_{q(\mathbf{s})}\log p(\mathbf{s}|\mathbf{e} = 0) - \mathbb{E}_{q(\mathbf{s})}\log p(\mathbf{s}|\mathbf{e} = 1) + \log\frac{p(\mathbf{e} = 0)}{p(\mathbf{e} = 1)}\right] \tag{31}$$

Adding and subtracting $\mathbb{E}_{q(\mathbf{s})}\log q(\mathbf{s})$ inside the exponent,

$$= \exp\left[D_{\mathrm{KL}}[q(\mathbf{s})||p(\mathbf{s}|\mathbf{e} = 1)] - D_{\mathrm{KL}}[q(\mathbf{s})||p(\mathbf{s}|\mathbf{e} = 0)] + \log\frac{p(\mathbf{e} = 0)}{p(\mathbf{e} = 1)}\right]. \tag{32}$$

Finally, we use the fact that $\sum_{\mathbf{e}\in\mathbf{E}} q(\mathbf{e}) = 1$ to get $q(\mathbf{e} = 0)$,

$$\frac{q(\mathbf{e} = 0)}{1 - q(\mathbf{e} = 0)} = \exp\left[D_{\mathrm{KL}}[q(\mathbf{s})||p(\mathbf{s}|\mathbf{e} = 1)] - D_{\mathrm{KL}}[q(\mathbf{s})||p(\mathbf{s}|\mathbf{e} = 0)] + \log\frac{p(\mathbf{e} = 0)}{p(\mathbf{e} = 1)}\right], \tag{33}$$

and solving for $q(\mathbf{e} = 0)$ yields,

$$q(\mathbf{e} = 0) = \frac{\exp\left[D_{\mathrm{KL}}[q(\mathbf{s})||p(\mathbf{s}|\mathbf{e} = 1)] - D_{\mathrm{KL}}[q(\mathbf{s})||p(\mathbf{s}|\mathbf{e} = 0)] + \log\frac{p(\mathbf{e}=0)}{p(\mathbf{e}=1)}\right]}{1 + \exp\left[D_{\mathrm{KL}}[q(\mathbf{s})||p(\mathbf{s}|\mathbf{e} = 1)] - D_{\mathrm{KL}}[q(\mathbf{s})||p(\mathbf{s}|\mathbf{e} = 0)] + \log\frac{p(\mathbf{e}=0)}{p(\mathbf{e}=1)}\right]}. \tag{34}$$

Equation 34 shows the formulation of distribution $q(\mathbf{e})$ in terms of the VaDE trick. We note that under the assumption that $p(\mathbf{e} = 0) = p(\mathbf{e} = 1) = 0.5$, the formulation is a softmax function of the difference between the KL divergences indicated in Eq. 4. As such, the proposed estimation of $q(\mathbf{e})$ may be seen as taking a sample from the most likely component in the VaDE-based distribution in Eq. 34, under the assumption of equal prior probabilities.

## C EXTRA RESULTS

### C.1 QUALITATIVE COMPARISONS

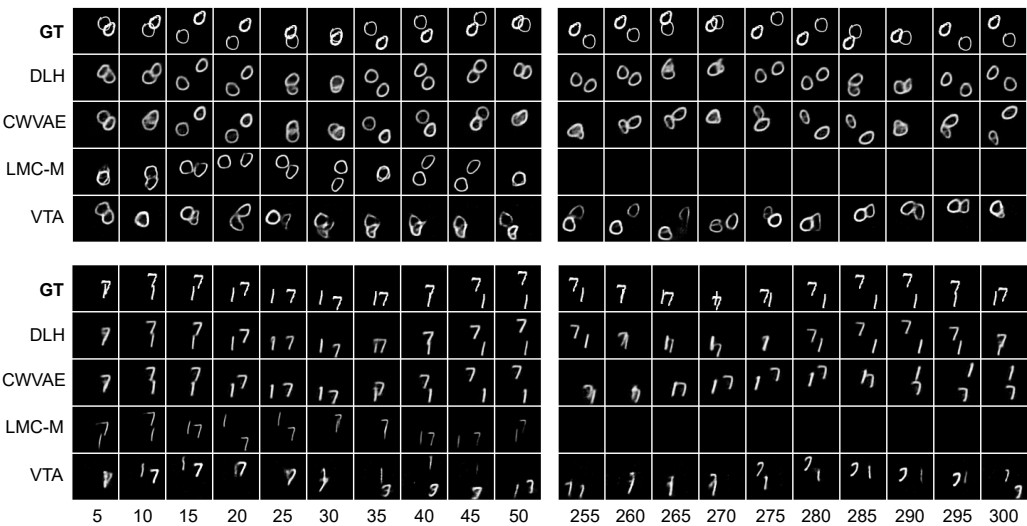

Figure 7: Qualitative comparisons of open-loop predictions using Moving MNIST. All models are given 30 context frames (36 for CW-VAE). The figures show the first 50 and last 45 timesteps in the produced rollouts. The context frames are not shown here. **GT** denotes the ground-truth sequence.

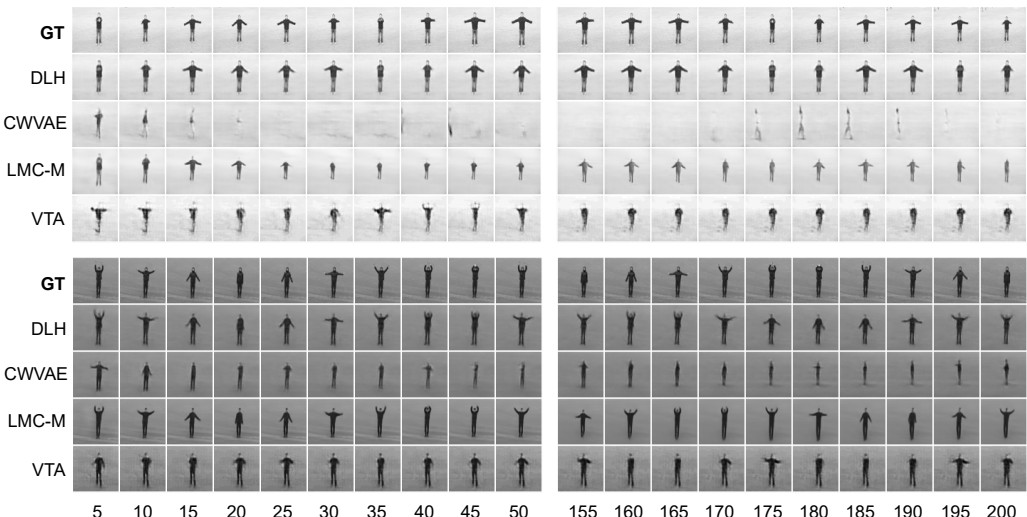

Figure 8: Qualitative comparisons of open-loop predictions using KTH. All models are given 30 context frames (36 for CW-VAE). The figures show the first 50 and last 45 timesteps in the produced rollouts. The context frames are not shown here. **GT** denotes the ground-truth sequence.

## C.2 PER-FRAME PREDICTION EVALUATION

Figure 9: Per-frame video prediction performance against the benchmarks on the Moving MNIST and KTH datasets. They y-axes correspond to the SSIM↑, PSNR↑, and LPIPS↓ metrics. The x-axes correspond to the number of open-loop prediction steps.

## C.3 SELECTION OF THE NUMBER OF LEVELS

We demonstrate in more detail the ability of DLH to converge its structure and employ only the necessary amount of resources in processing data. In Section 2.3, using the Moving Ball dataset, we showed that irrespective of the number of levels the model possesses, it only partially uses its latent hierarchy. In particular, we reported two metrics: the average number of levels used by the model and the average total KL divergence. For both metrics, we reported similar values across the instances of DLH with different total number of levels.

Table 5: Per-level KL divergences of DLH (Moving Ball dataset)

| Levels | KL (L1) | KL (L2) | KL (L3) | KL (L4) | KL (L5) |
|:------:|:-------:|:-------:|:-------:|:-------:|:-------:|
| 2 | 35.1 | 6.8 | - | - | - |
| 3 | 34.7 | 7.8 | **0.3** | - | - |
| 4 | 34.3 | 6.5 | **0.2** | **0.1** | - |
| 5 | 36.1 | 6.6 | **0.1** | **0.1** | **0.2** |

Notably, the KL divergence is often used as a measure of the amount of information stored in the latent variables of the model. By breaking down the contributions of each latent level to the total value of the KL loss, we can get a glimpse into how DLH employs the different hierarchical levels. Table 5 shows the average per-level KL loss of DLH. As can be observed, levels $> 2$ are significantly lower, suggesting that no information is being stored in those levels, as the model naturally 'collapses' them. More visually, we can sample from these low-KL levels ($> 2$), while keeping high-KL levels fixed ($\leq 2$), and vice versa, in order to see if these levels contribute to the variations in the reconstructed images. Figures 10 and 11 show that, indeed, samples from the low-KL levels exhibit minimum variations, while high-KL levels seem to encode most of the important information. These results once again illustrate how DLH naturally tends to use the minimum amount of resources necessary for modelling the data. Furthermore, this property hints at a potential method for selecting the appropriate number of levels – by monitoring the values of the KL for models with different number of hierarchical levels.

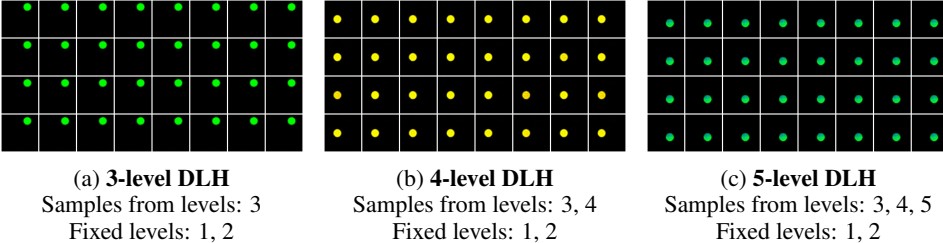

(a) **3-level DLH**
Samples from levels: 3
Fixed levels: 1, 2

(b) **4-level DLH**
Samples from levels: 3, 4
Fixed levels: 1, 2

(c) **5-level DLH**
Samples from levels: 3, 4, 5
Fixed levels: 1, 2

Figure 10: **Samples from levels** $> 2$. As can be seen, there are no variations in the observed reconstructions, indicating that no information is stored in these levels. This is corroborated by the low values of the KL divergence associated with the hierarchical levels.

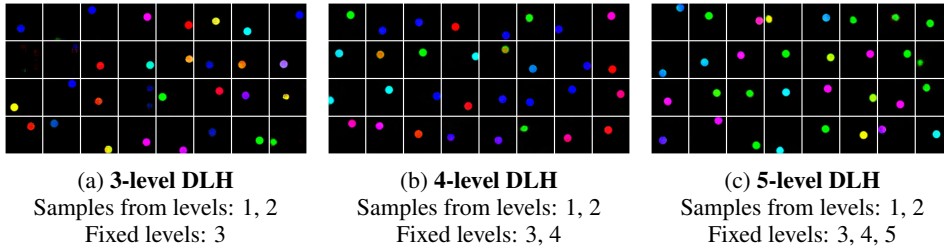

(a) **3-level DLH**
Samples from levels: 1, 2
Fixed levels: 3

(b) **4-level DLH**
Samples from levels: 1, 2
Fixed levels: 3, 4

(c) **5-level DLH**
Samples from levels: 1, 2
Fixed levels: 3, 4, 5

Figure 11: **Samples from levels** $\leq 2$. In contrast to samples in Figure 10, we now observe high variation in the reconstructed images, indicating that the data is primarily modelled by the bottom two levels. This is in line with the high values of the KL divergences associated with these levels.

