# OpenReview forum: "Long-horizon video prediction using a dynamic latent hierarchy"
_ICLR.cc/2023/Conference — Submitted to ICLR 2023_

### Official Review · Reviewer_UWJy · 2022-10-22

**Confidence:** 3
**Correctness:** 2
**Technical Novelty And Significance:** 2
**Empirical Novelty And Significance:** 2
**Recommendation:** 3

**Clarity, Quality, Novelty And Reproducibility:**

**Novelty**

[Extremely Major:] The paper heavily relies on Variational Predictive Routing [Zakharov et al, 2021] in several places, but the paper does not describe what exactly is novel over the VPR setting. More specifically, what are the changes or adaptations made to the VPR model in this paper? In my opinion, the paper should relegate the descriptions of the VPR model and focus only on their contributions in the Dynamic Latent Hierarchy section. The paper should also offer comparisons to VPR as a baseline, showing that their proposed modifications lead to superior performance.

**Clarity and Quality**

The algorithm is somewhat confusing partially because in some locations, the dependencies are omitted, and in others, they are not. I have posted my understanding of the algorithm from reading it a few times, with some questions. Updating the draft, in response to these questions, can improve the clarity of the paper.

* First, the posterior over the latent variables $q(s_t^n)$ is inferred in Eq 8b). From Section 2.2, apparently there is a dependency across levels in a top-down fashion, from higher to lower levels. I assume there is also a dependency across timesteps? Please update Eq 8b) to reflect both these dependencies.
* At every level and timestep, the algorithm compares the KL between the posterior $q(s_t^n | s_t^{>n}, x_t)$ and $P(s_{t-1}^n)$ with the KL between KL between the posterior $q(s_t^n | s_t^{>n}, x_t)$ and $P_{\theta}(s_t^n | s_{<t}^n)$. If the first term is higher, then the inferred latent variable $q(e_t^n = 1)$ or 0 otherwise. Is this the same prior as in Eq 8b) or a different prior? Are these also the dependencies across levels in this prior?
* Is the prior over the latent emission variables conditioned on the previous emission variables? Equation 1 describes that $e_t^n$ is dependent on $e_t^{<n}$ while eq 8c) omits these dependencies. If it is conditioned on the sampled emission variables, then how is backpropagation done through this discrete sampling operation, while minimizing the ELBO?
* In Section 2.3, if $e_t^{n-1}=0$, then $e_t^{n}$ is also set to 0. Also, latent discrete variable, $q(s_t^n)$ is set to the $q(s_{t-1}^n)$. However, since $q(s^{n+1})$ was conditioned on the previous value of $q(s_t^n)$? Is this unchanged or kept the same? What is the rationale behind this?
* Figure 2a is confusing because there are no superscripts. Superscripts are necessary to show the evolution of the states over time.
* The dependencies across $x$, $c$, $d$ and $s$ should be made very clear in Section 2.5. For example. how does $s$ depend on the temporal state?
* Is $x_t^0$ the image and $c_t^0$ correspond to the output image?
* Are the parameters shared across all levels? Since each level $l$ has only dependencies to the levels above them, how is this "variable length" in terms of levels handled?

**Reproducibility**
It would be nice if some pesudocode of the training loop is provided. It would make it easier to digest and understand.

**Experiments**

* How were the hyperparameters of DHL tuned, for example, the number of levels.? On which validation set and on which metric?
* How does the computational efficiency and number of parameters compare to the baselines?
* Could the authors also plot the per-frame LPIPS, SSIM? It would also be nice to report the FVD Scores? What are the star superscripts in Table 1?
* Figure 5 demonstrates reconstructed frames retrieved by sampling the different hierarchical levels of the model. What does this exactly mean? Given a video, first the posterior latent distribution is inferred and then this is replaced with sampling from the prior. Is that correct?
* In Figure 5, on the bottom most subplot, it makes sense that L1 is set to 1, since the authors explicitly set it to be 1 in Section 2.3. So its not super meaningful.
* What is average hierarchical depth and how is it computed?
* Table 3 specifies different levels but not which specific level.


**Strength And Weaknesses:**

**Strengths**

The problem is well motivated as stated in the introduction. In a video, it is expected that latent variables operate at different timescales. Providing this as an inductive bias in video modeling, could help in video prediction and could lead to interesting insights.

**Weaknesses**

I have some concerns with the novelty, clarity and experiments in the paper. See below for detailed feedback and for actionable suggestions.


**Summary Of The Paper:**

This paper proposes Dynamic Latent Hierarchy, a latent-variable video prediction model. This model can learn a hierarchy of latent variables where the latent variable at each level of the hierarchy, operates at a different timescale. Unlike related work, ClockWork VAE’s, (Saxena et al. 2021), whether or not a latent variable at a certain level is activated is not hard coded. Instead, the paper proposes to cast this as a Bernoulli latent variable and learn it using variational inference.

Improvements are shown on KTH, DML Mazes and Moving Mnist on SSIM and PSNR over 3 competing methods. Qualitiative experiments on KTH show that the higher levels get activated lesser as compared to the lower levels. Similarly, on a synthetic moving ball dataset, when the color is changed stochastically, on higher levels of stochasticity, the emission probability is higher.


**Summary Of The Review:**

I've provided detailed feedback above with respect to my concerns regarding the paper. I'm open to adjusting my rating if authors can clarify my concerns.

---

> ### Author Response · Authors · 2022-11-20
> **Response to Reviewer UWJy (Part 1)**
>
> Dear Reviewer UWJy,
>
> Thank you for your insightful feedback on our paper. Please let us address some of the questions you raised in your review.
>
> ---
>
> **Q:** *The paper heavily relies on Variational Predictive Routing [Zakharov et al, 2021] in several places, but the paper does not describe what exactly is novel over the VPR setting. More specifically, what are the changes or adaptations made to the VPR model in this paper? In my opinion, the paper should relegate the descriptions of the VPR model and focus only on their contributions in the Dynamic Latent Hierarchy section. The paper should also offer comparisons to VPR as a baseline, showing that their proposed modifications lead to superior performance.*
>
> **A:** Thank you for this comment.
>
> We were indeed inspired by the prior work on the VPR model (Zakharov et al. 2022), which showed an architecture of a generative model with appealing representation learning capabilities, including against VTA (Kim et al. 2019). That being said, there are a number of important differences and technical contributions of our work against VPR. We hope that the explanations below will convince you that they hold significant weight. Where appropriate, we will reference any amendments made to the manuscript.
>
> You have rightfully pointed out the close connection of DLH with VPR -- we had no intention to hide these similarities and made sure that they are exposed to the reader. But, it seems, in doing so, we have failed to properly emphasise the contributions of our work and erroneously attributed part of it to Zakharov et al., thereby stirring some confusion that we wish to resolve now.
>
> In designing DLH, we wanted to combine the reported benefits of VPR (non-parametric state change detector and meaningful latent spaces) with the relevant ideas from indicator-based models capable of video prediction. Although DLH may seem like an extension to the VPR model, we would like to emphasise that *it is absolutely not*.
>
> First, it is important to note that VPR *is not* a video prediction model. Its contributions were only related to representation learning, time-agnostic prediction accuracy, and the *non-parametric* approach for detecting hierarchical state updates in an online fashion. VPR cannot be benchmarked on video prediction alongside the models presented in this paper, and its value was almost exclusively connected to the ability of building disentangled hierarchical representations.
>
> The fundamental question we concerned ourselves with was: can we introduce the indicator variable (that would allow for video prediction) into the graphical model of a hierarchical temporal VAE, while retaining the online non-parametric estimation ability? In the process of working on DLH, we found that the answer to this question lies in explicitly formulating the latent state as a *temporal Mixture of Gaussians*, which: (1) provides the ground to derive a method for non-parametric estimation of the indicator variable based on the theoretical results found in the GMM-VAE literature (*as opposed to the heuristic-based technique in VPR*), but equally (2) results in a substantially different graphical model and objective function.
>
> These changes resulted in a model that, although conceptually similar to the class of models mentioned in the paper (CW-VAE, VPR), is significantly different in a number of ways. Because of this, we felt it would be inappropriate to frame DLH as an extension of VPR.
>
> Let us now succinctly summarise the important differences between DLH and VPR, specifically.
>
> **Video prediction**. First, as mentioned, VPR *is not* a video prediction model, as it does not model the temporal evolution of states with respect to the physical timescale of the dataset.
>
> [to be continued]

---

> > ### Author Response · Authors · 2022-11-20
> > **Response to Reviewer UWJy (Part 2)**
> >
> > **Different graphical models and objective functions.** Second, by formulating the latent state as a *temporal Mixture of Gaussians* with the concomitant discrete latent indicator variable $e$, we have effectively arrived at a different graphical model and, by extension, a different objective function (ELBO). This introduces an interesting distinction between the objective functions of VPR (and CW-VAE) to the objective function of DLH. In particular, apart from the KL divergence over the indicator variable in Eq.8c, the major difference can be seen in Eq.8b (and more thoroughly explained in Eq.9). Specifically, the KL divergence over the posterior $q(s)$ and prior latent states $p(s|e)$ is taken with the expectation over the posterior $q(e)$. As a result, depending on the inferred distribution $q(e)$, the posterior will be forced either closer to the *static* $p(s|e=0)$ or to the *change* $p(s|e=1)$ component of the prior MoG distribution. As explained in the paper, this allows for a natural organisation of the latent space, as temporally-persistent states get clustered together and occupy similar parts of the latent space. In contrast, in VPR (and CW-VAE), there is no such property -- VPR only learns to transition between states that were detected to be different in the latent space via the 'event detection criteria'.
> >
> > **Different non-parametric estimation.** We have certainly caused some confusion when attempting to appeal to conceptual similarities between VPR's method of detecting events and DLH's inference procedure of the indicator variable. In reality, VPR's non-parametric technique for detecting state changes is substantially different from the method of approximating posterior $q(e)$ in DLH. In particular, in VPR, it consists of two criteria that were designed to capture predictable (CE criterion) and unpredictable (CU criterion) changes in the latent states. While the CU criterion has absolutely no relation to the method of estimating of $q(e)$ in DLH, the CE criterion has some conceptual similarities to our technique, but is not mathematically equivalent. Moreover, VPR's technique is heuristically-motivated, and does not involve any theoretical justification with respect to the ELBO maximisation objective. Zakharov et al. similarly do not consider the CE criterion as a likelihood ratio test; indeed, this interpretation is only valid under a particular assumption. In contrast, DLH's $q(e)$ approximation deals with a principled problem of inference and draws on the GMM-VAE literature, as opposed to the VPR's detection mechanism; specifically, the VaDE trick (Jiang et al., 2017; Falck et al., 2021). Applying this methodology in the context of temporal hierarchical VAEs is a novelty, and the more relevant connection to the VaED trick has been outlined in Appendix B.3. To summarise, in framing VPR's event detection method as being 'similar' to our inference procedure, we have erroneously misled you to believe that we are simply using the CE criterion presented in Zakharov et al., 2022. In reality, this was an attempt to appeal to: (1) conceptual similarity, (2) the non-parametric nature of the technique, and (3) the context in which it is being used (i.e. hierarchical temporal models). We have removed the misleading note in Section 2.2, leaving only the relevant connection to the VaDE trick.
> >
> > We hope that this explanation was sufficient in convincing you of the novelty of our work and of the significant differences between DLH and VPR. Please let us know if we could address your concerns.
> >
> > ---
> >
> > **Q:** *First, the posterior over the latent variables $q(s^n_t)$ is inferred in Eq 8b). From Section 2.2, apparently there is a dependency across levels in a top-down fashion, from higher to lower levels. I assume there is also a dependency across timesteps? Please update Eq 8b) to reflect both these dependencies.*
> >
> > **A:** Thank you for this suggestion. We have now updated the equations to reflect these dependencies.
> >
> > ---
> >
> > **Q:** *At every level and timestep, the algorithm compares the KL between the posterior $q(s^n_t | s_t^{>n}, x_t)$ and $p(s^n_{t-1})$ with the KL between the posterior $q(s^n_t | s_t^{>n}, x_t)$ and $p_\theta(s^n_{t} | s^n_{<t})$ . If the first term is higher, then the inferred latent variable $q(e^n_t=1)$ or 0 otherwise. Is this the same prior as in Eq 8b) or a different prior? Are these also the dependencies across levels in this prior?*
> >
> > **A:** It is indeed the same prior. In explaining the inference process over the indicator variable $e$ , we wanted to simplify the notation by omitting the conditional dependencies. However, we recognise that the way in which it was done may have made it more confusing. Therefore, we have now introduced a small clarification immediately after Eq. 3. We hope that this resolves the ambiguity.

---

> > > ### Author Response · Authors · 2022-11-20
> > > **Response to Reviewer UWJy (Part 3)**
> > >
> > > **Q:** *Is the prior over the latent emission variables conditioned on the previous emission variables? Equation 1 describes that $e^n_t$ is dependent on $e^{n}_{<t}$ while eq 8c) omits these dependencies. If it is conditioned on the sampled emission variables, then how is backpropagation done through this discrete sampling operation, while minimizing the ELBO?*
> > >
> > > **A:** Thank you for pointing this out. This was indeed a typo. The prior indicator model is not conditioned on the previous discrete MoG component variables. This has now been fixed.
> > >
> > > ---
> > >
> > > **Q:** *In Section 2.3, if $e_t^{n-1} = 0$ , then $e^n_t$ is also set to 0. Also, latent discrete variable, $q(s^n_t)$ is set to the $q(s^n_{t-1})$ . However, since $q(s^{n+1})$ was conditioned on the previous value of $q(s^{n}_t)$ ? Is this unchanged or kept the same?*
> > >
> > > **A:** This is a great question! The inference procedure consists of two sequential steps: (1) inferring the latent indicator variables $q(e^n_t)$ , and (2) inferring the posterior states $q(s^n_t|\cdot)$ . But how can we approximate indicator variables before inferring all of the hierarchical posteriors? The answer lies in the nested timescales approximation (Eq.5), which was done to improve the computational efficiency of the model. In particular, when computing the indicator variables using the expected log-likelihood ratio at level $n$ , we assume that the hierarchical contexts (i.e. $q(s_t^{>n})$ ) have not changed, meaning $q(s_t^{>n}) \leftarrow q(s_{t-1}^{>n}) = p(s_t^{>n} | e_t^{>n} = 0)$ . The expected log-likelihood ratio is therefore computed under the expectation of static hierarchical priors, rather than the newly-inferred posteriors.
> > >
> > > Why is this important? Because it allows us to perform inference *only* up to some level $k$ at which $e^k_t = 0$ , after which we can apply the approximating assumption from Eq.5. This means we do not need to compute the posteriors $q(s^{>k}_t)$. To make this point absolutely clear, we have added a detailed explanation in Appendix B.2 and a pseudocode in Appendix A.2.
> > >
> > > ---
> > >
> > > **Q:** *Figure 2a is confusing because there are no superscripts. Superscripts are necessary to show the evolution of the states over time.*
> > >
> > > **A:** Thank you for this suggestion. We have now updated the figure.
> > >
> > > ---
> > >
> > > **Q:** *The dependencies across $x, c, d$ and $s$ should be made very clear in Section 2.5. For example. how does $s$ depend on the temporal state?*
> > >
> > > **A:** Thank you. We have now made these dependencies explicit in Equations 10-15.
> > >
> > > ---
> > >
> > > **Q:** *Is the $x^0_t$ image and $c^0_t$ correspond to the output image?*
> > >
> > > **A:** You are correct. We have amended Sec 2.5 to make this clear.
> > >
> > > ---
> > >
> > > **Q:** *Are the parameters shared across all levels? Since each level $l$ has only dependencies to the levels above them, how is this "variable length" in terms of levels handled?*
> > >
> > > **A:** The parameters are not shared across the levels. With regards to the second question, could you please clarify what you mean exactly?
> > >
> > > ---
> > >
> > > **Q:** *It would be nice if some pesudocode of the training loop is provided. It would make it easier to digest and understand.*
> > >
> > > **A:** Great suggestion. We have included the pseudocode of DLH in Appendix A.2.
> > >
> > > ---
> > >
> > > **Q:** *How were the hyperparameters of DHL tuned, for example, the number of levels.? On which validation set and on which metric?*
> > >
> > > **A:** We have done very minimal hyperparameter tuning. A small subset of the test data (10%) has been used as the validation set. Tuning was done primarily for the standard deviation of the Gaussian reconstruction model (see Appendix A.1) using the SSIM metric and the qualitative reconstruction and prediction results.
> > >
> > > In choosing the number of levels, we have taken into account prior work on scalable temporal VAEs; in particular, CW-VAE. To make the comparisons fair, we have therefore decided to employ the same number of levels. Additionally, however, we have added an interesting discussion regarding the process of selecting the number of levels in Appendix C.2, which relates to the results reported in Section 4.3.
> > >
> > > ---
> > >
> > > **Q:** *Number of parameters*
> > >
> > > **A:** In terms of the parameters, our model is the second smallest model out of the used baselines. In particular, **DLH**: 7M, **LMC-Memory**: 34M, **CW-VAE**: 12M, **VTA**: 3M. We believe this is yet another indicator of the effectiveness of our model. We have included these statistics in Table 1.

---

> > > > ### Author Response · Authors · 2022-11-20
> > > > **Response to Reviewer UWJy (Part 4)**
> > > >
> > > > **Q:** *Could the authors also plot the per-frame LPIPS, SSIM? It would also be nice to report the FVD Scores?*
> > > >
> > > > **A:** Thank you for this suggestion. We have added the per-frame SSIM, PSNR, and LPIPS plots for Moving MNIST and KTH datasets in Appendix C.2. Unfortunately, we did not have enough time to implement the FVD scores.
> > > >
> > > > ---
> > > >
> > > > **Q:** *What are the star superscripts in Table 1?*
> > > >
> > > > **A:** The stars denote <5% standard deviation of the reported metrics; we forgot to include this description in the previous version of the manuscript. This has now been fixed.
> > > >
> > > > ---
> > > >
> > > > **Q:** *Figure 5 demonstrates reconstructed frames retrieved by sampling the different hierarchical levels of the model. What does this exactly mean? Given a video, first the posterior latent distribution is inferred and then this is replaced with sampling from the prior. Is that correct?*
> > > >
> > > > **A:** To produce these samples, we follow a simple procedure. (1) initialise a pre-trained model, (2) make a one-step prediction using empty temporal contexts ($d^n_{0}=\mathbf{0}$), (3) produce latent states corresponding to the prediction $p_\theta(s^n_1 | d^n_1, c^n_1)$, and finally (4) sample multiple times from $p(s^n_1)$ for some level $n$ of interest, while keeping all other level samples fixed. This way, we are not probing variations in the model's predictions given a video, but rather force it to first predict the most general prior (since the model is given no context frames) and then inspect what information is being stored in the levels via sampling.
> > > >
> > > >
> > > > ---
> > > >
> > > > **Q:** *In Figure 5, on the bottom most subplot, it makes sense that L1 is set to 1, since the authors explicitly set it to be 1 in Section 2.3. So its not super meaningful.*
> > > >
> > > > **A:** Thank you for this comment. We certainly agree that it is not particularly interesting; however, we felt it is important, as it graphically illustrates an example of the DLH's overall temporal structure. Excluding the first level may have caused confusion.
> > > >
> > > > ---
> > > >
> > > > **Q:** *What is average hierarchical depth and how is it computed?*
> > > >
> > > > **A:** Average hierarchical depth is computed by simply taking the sum of the MoG components, $e^n_t$. Recall that $e^n_t = 0$ corresponds to the static component, while $e^n_t = 1$ to the change component. As such, we can compute the average hierarchical depth using $\bar{L} = \frac{1}{T} \sum_{t=1}^T \sum_{n=1}^N e^n_t$. We have now added this clarification in Section 4.3.
> > > >
> > > > ---
> > > >
> > > > **Q:** *Table 3 specifies different levels but not which specific level.*
> > > >
> > > > **A:** We are not exactly sure what you mean by this comment. If you mean the levels of stochasticity, they are specified in the table using $\lambda$. If you mean the level for which the average value of $p_\theta(e^n)$ is calculated, it is level two. We have now added some clarifications with regards to this.
> > > >
> > > > ---
> > > >
> > > > Once again, thank you so much for your review. We have found it very insightful for improving our manuscript. Please let us know if we could address your concerns -- we would be very happy to hear back from you!

---

### Official Review · Reviewer_1qre · 2022-10-24

**Confidence:** 5
**Correctness:** 3
**Technical Novelty And Significance:** 2
**Empirical Novelty And Significance:** 3
**Recommendation:** 3

**Clarity, Quality, Novelty And Reproducibility:**

This paper is clearly written but has insufficient novelty for the proposed method. Please see the comments above for more details.

**Strength And Weaknesses:**

Strength:
1. This paper is clearly written and easy to follow.
2. By considering modular representations, this paper has proposed a simple yet effective hierarchical framework to model the mixed spatiotemporal dynamics in video sequence at different timescales, and also achieves great quantitative results.
3. Detailed validations are performed on multiple datasets. In addition to the significant improvement in SSIM/PSNR, in Fig 5, the authors also tried to demonstrate the advantages of the proposed method in decoupling different spatiotemporal dynamics from the qualitative visualization.

Weaknesses:
1. My main concern lies in the novelty of the proposed approach, which seems to be a simple extension of the hierarchical model from Zakharov et al. (2022). If I understand correctly, the biggest difference between them is the binary indicator e_t^n, which is used to detect event (change or static). However, similar techniques has been well explored by Kim et al. (2019). The technical contributions of the model are not convincing to me and require further clarifications.
2. I also have some doubts about the connections between the example explained in Fig 1 and the method of "nested timescales" proposed in Sec 2.3. I understand that the authors attempted to describe the hierarchical state organization, but in Fig 1, it seems that the dynamics of the panda has nothing to do with that of the airplane. Therefore, if Fig 1 correctly states the motivation of the method, it should learn the dynamics at different timescales in parallel state transition branches, instead of the hierarchical one. In other words, in Eq (5), the indicator of state changes at layer n (e_t^n=0) should not be completely determined by e_t^{n-1}=0.
3. Although in Sec 2.3, the method is claimed to reduce the computational complexity, no corresponding empirical evidence was given in Sec 4 that could support the efficiency of the proposed model.
4. In my view, the numbers of state levels that are used to model complex scenes (such as KTH or even datasets with more complex visual dynamics) and simple scenes (such as Moving MNIST) may be significantly different. How should the hierarchies be determined for different scenarios? In other words, how many levels of latent states should be selected for different datasets? As shown in Table 1 and Fig 5 for selecting 3 levels throughout the first three datasets, can the conclusions be extended to general scenarios? If not, can the authors provide some reference schemes for selecting the number of levels. Besides, it would improve the fairness of the comparison to implement a VTA model that also employs a 3-level architecture with similar number of parameters.
5. What does * indicate in Table 1?
6. In Figs 4-5, I strongly encourage the authors to include more qualitative comparisons for long-term video prediction with existing benchmarks. The proposed method is expected to generate future frames with higher visualization quality. In addition, it would be nice if the authors could give clearer textual explanations in Figs 4-5. I am confused on what each image sequence represents...Are those in the first row the ground-truth images, or does each row represent a different prediction sequence by the model based on the same input sequence?

Other suggestions (NOT weaknesses):
1. If the experimental conditions allow, it is essential to show the model performance on more real-world datasets with large spatiotemporal uncertainty, such as RoboNet or BAIR. The authors may consider the action-conditioned video prediction setup because it would be interesting to analyze the correspondence between the learned indicators and the given action signals.
2. In addition to using the best prediction among 100 prediction samples, the authors may evaluate the models using the average performance of the 100 samples as well as the worst cases.


**Summary Of The Paper:**

This paper presents a new video prediction model in the form of neural networks, which combines the advantages of hierarchical clockwork models (from CW-VAE) and jumpy recurrent modeling controlled by binary indicators (from VTA).
- Major: It provides a novel graphical model for spatiotemporal prediction. To avoid the unnecessary accumulation of prediction errors and improve computational efficiency, the authors propose to disentangle the temporal dynamics into hierarchical latent states and predict whether the states would change or remain static at each future timestep.
- Minor: The proposed model is shown effective on four datasets, improving previous benchmarks including CW-VAE and VTA.

**Summary Of The Review:**

Overall, it is a good paper that improves the existing benchmarks on multiple video prediction datasets. However, it needs significant improvement, specifically in the following aspects:
1. The technical contribution of the paper. Although the proposed approach is new in terms of the graphical model shown in Fig 1, the key insight of combining hierarchical latent representations and the jumpy state transitions for modeling video dynamics adaptively at different timescales has been explored by existing work with similar techniques (using binary indicators for the state changes).
2. More explanation of the connection between the motivations shown in Fig 1 and the proposed approach.
3. Additional experiments on model efficiency and more qualitative comparisons for long-term prediction should be included to understand the effectiveness of the approach.

---

> ### Author Response · Authors · 2022-11-20
> **Response to Reviewer 1qre (Part 1)**
>
> Dear Reviewer 1qre,
>
> Thank you for your review of our paper! In what follows, we wish to address some of the questions you raised.
>
> **Q:** *My main concern lies in the novelty of the proposed approach, which seems to be a simple extension of the hierarchical model from Zakharov et al. (2022). If I understand correctly, the biggest difference between them is the binary indicator et^n, which is used to detect event (change or static). However, similar techniques has been well explored by Kim et al. (2019). The technical contributions of the model are not convincing to me and require further clarifications.*
>
> **A:** Thank you for this comment. We were indeed inspired by the prior work on the VPR model (Zakharov et al. 2022), which showed an architecture of a generative model with appealing representation learning capabilities, including against VTA (Kim et al. 2019). That being said, there are a number of important differences and technical contributions of our work against VPR. We hope that the explanations below will convince you that they hold significant weight. Where appropriate, we will reference any amendments made to the manuscript.
>
> You have rightfully pointed out the close connection of DLH with VPR -- we had no intention to hide these similarities and, indeed, made sure that they are exposed to the reader. But, it seems, in doing so, we have failed to properly emphasise the contributions of our work and erroneously attributed part of it to Zakharov et al., thereby stirring some confusion that we wish to resolve now.
>
> In designing DLH, we wanted to combine the reported benefits of VPR (non-parametric state change detector and meaningful latent spaces) with the relevant ideas from indicator-based models capable of video prediction. Although DLH may seem like an extension to the VPR model, we would like to emphasise that *it is absolutely not*.
>
> First, it is important to note that VPR *is not* a video prediction model. Its contributions were only related to representation learning, time-agnostic prediction accuracy, and the *non-parametric* approach for detecting hierarchical state updates in an online fashion. VPR cannot be benchmarked on video prediction alongside the models presented in this paper, and its value was almost exclusively connected to the ability of building disentangled hierarchical representations.  At the same time, VTA relies on a parametric estimation of the indicator variable and is thus capable of video prediction; however, its performance was inferior to VPR in spatiotemporal structure discovery, which was in detail analysed in Zakharov et al. 2022.
>
> As such, the fundamental question we concerned ourselves with was: can we introduce the indicator variable (that would allow for video prediction) into the graphical model of a hierarchical temporal VAE, while retaining the online non-parametric estimation ability? In the process of working on DLH, we found that the answer to this question lies in explicitly formulating the latent state as a *temporal Mixture of Gaussians*, which (1) provides the ground to derive a method for non-parametric estimation of the indicator variable based on the theoretical results found in the GMM-VAE literature (*as opposed to the heuristic-based technique in VPR*), but equally (2) results in a substantially different graphical model and objective function.
>
> Let us now succinctly summarise the important differences between DLH and VPR, specifically.
>
> **Video prediction**. VPR *is not* a video prediction model, as it does not model the temporal evolution of states with respect to the physical timescale of the dataset.
>
> **Different graphical models and objective functions.** By formulating the latent state as a *temporal Mixture of Gaussians* with the concomitant discrete latent indicator variable $e$, we have effectively arrived at a different graphical model and, by extension, a different objective function (ELBO). This introduces an interesting distinction between the objective functions of VPR (and CW-VAE) to the objective function of DLH. In particular, apart from the KL divergence over the indicator variable in Eq.8c, the major difference can be seen in Eq.8b (and more thoroughly explained in Eq.9). Specifically, the KL divergence over the posterior $q(s)$ and prior latent states $p(s|e)$ is taken with the expectation over the posterior $q(e)$. As a result, depending on the inferred distribution $q(e)$, the posterior will be forced either closer to the *static* $p(s|e=0)$ or to the *change* $p(s|e=1)$ component of the prior MoG distribution. As explained in the paper, this allows for a natural organisation of the latent space, as temporally-persistent states get clustered together and occupy similar parts of the latent space. In contrast, in VPR (and CW-VAE), there is no such property -- VPR only learns to transition between states that were detected to be different in the latent space via the 'event detection criteria'. [To be continued]

---

> > ### Author Response · Authors · 2022-11-20
> > **Response to Reviewer 1qre (Part 2)**
> >
> > **Different non-parametric estimation.** We have certainly caused some confusion when attempting to appeal to conceptual similarities between VPR's method of detecting events and DLH's inference procedure of the indicator variable. In reality, VPR's non-parametric technique for detecting state changes is substantially different from the method of approximating posterior $q(e)$ in DLH. In particular, in VPR, it consists of two criteria that were designed to capture predictable (CE criterion) and unpredictable (CU criterion) changes in the latent states. While the CU criterion has absolutely no relation to the method of estimating of $q(e)$ in DLH, the CE criterion has some conceptual similarities to our technique, but is not mathematically equivalent. Moreover, VPR's technique is heuristically-motivated, and does not involve any theoretical justification with respect to the ELBO maximisation objective. Zakharov et al. similarly do not consider the CE criterion as a likelihood ratio test; indeed, this interpretation is only valid under a particular assumption. In contrast, DLH's $q(e)$ approximation deals with a principled problem of inference and draws on the GMM-VAE literature, as opposed to the VPR's detection mechanism; specifically, the VaDE trick (Jiang et al., 2017; Falck et al., 2021). Applying this methodology in the context of temporal hierarchical VAEs is a novelty, and the more relevant connection to the VaED trick has been outlined in Appendix B.3. To summarise, in framing VPR's event detection method as being 'similar' to our inference procedure, we have erroneously misled you to believe that we are simply using the CE criterion presented in Zakharov et al., 2022. In reality, this was an attempt to appeal to: (1) conceptual similarity, (2) the non-parametric nature of the technique, and (3) the context in which it is being used (i.e. hierarchical temporal models). We have removed the misleading note in Section 2.2, leaving only the relevant connection to the VaDE trick.
> >
> >
> > To summarise, the main novelty of our work comes from formulation of a class of hierarchical temporal VAEs in terms of the temporal Mixtures of Gaussians. This opened the door to the non-parametric estimation of the 'indicator variables' (mixture components) found in the GM-VAE literature, and to the objective function that encourages spatiotemporal representation learning (Eq.9).
> >
> > ---
> >
> > **Q:** *I also have some doubts about the connections between the example explained in Fig 1 and the method of "nested timescales" proposed in Sec 2.3. I understand that the authors attempted to describe the hierarchical state organization, but in Fig 1, it seems that the dynamics of the panda has nothing to do with that of the airplane. Therefore, if Fig 1 correctly states the motivation of the method, it should learn the dynamics at different timescales in parallel state transition branches, instead of the hierarchical one. In other words, in Eq (5), the indicator of state changes at layer n (et^n=0) should not be completely determined by et^{ n-1}=0.*
> >
> > **A:** This is a great note. Indeed, this figure was primarily displayed to introduce the concept of modelling a video sequence as a collection of features that possess different temporal dynamics, rather than relate to the assumption of nested timescales. This assumption is introduced later, and is, in fact, not central to our work, as it only provides a useful inductive bias for *enforcing* a particular kind of generative structure.  The figure, on the other hand, deals with a more general conceptualisation of video modelling, for which the assumption of nested timescales is not a requirement (we also mention it in Sec. 2.3).
> >
> > ---
> >
> > **Q:** *Although in Sec 2.3, the method is claimed to reduce the computational complexity, no corresponding empirical evidence was given in Sec 4 that could support the efficiency of the proposed model.*
> >
> > **A:** Thank you for this comment. The reduction in the computational complexity arises as a result of blocking inference beyond the level at which the static mixture component is inferred (Eq. 5) -- hence the claimed computational efficiency. Without this approximation, DLH would need to infer *all* hierarchical posteriors at every timestep. Note that we did not claim better computational efficiency against our benchmarks -- the point relates only to the version of DLH without the proposed assumption, which is why we were hesitant to include any empirical results. Nevertheless, in order to make this point more clear, we have added Appendix B.2, which explains how we arrive at these computational savings.

---

> > > ### Author Response · Authors · 2022-11-20
> > > **Response to Reviewer 1qre (Part 3)**
> > >
> > > **Q:** *In my view, the numbers of state levels that are used to model complex scenes (such as KTH or even datasets with more complex visual dynamics) and simple scenes (such as Moving MNIST) may be significantly different. How should the hierarchies be determined for different scenarios? In other words, how many levels of latent states should be selected for different datasets? As shown in Table 1 and Fig 5 for selecting 3 levels throughout the first three datasets, can the conclusions be extended to general scenarios? If not, can the authors provide some reference schemes for selecting the number of levels. Besides, it would improve the fairness of the comparison to implement a VTA model that also employs a 3-level architecture with similar number of parameters.*
> > >
> > > **A:** This is a brilliant question, and it really speaks to some of the central ideas behind our work. In particular, DLH is based on the concept of *dynamic manipulation* of the latent structure (hierarchical and temporal) by means of the temporal Gaussian mixture. This allows the model to flexibly control the amount of hierarchical resources employed for processing a video. By resources, we really mean the number of levels that are engaged in the inference and prediction procedure, which is why we find the results in Sec. 4.3 so insightful -- these provide the essential intuition for answering your question. In particular, using the simplistic Moving Ball dataset, we have shown that regardless of the total number of hierarchical levels that DLH possesses, the property of the dynamic hierarchy allows the model to converge to the similar *average* number of levels it employs for processing. In effect, this corresponds to the simplification of the DLH's structural complexity in response to a dynamically and visually uncomplicated dataset. This point suggests an important corollary: the model *automatically* adjusts its *effective* hierarchical depth.
> > >
> > > With this in mind, we propose that choosing the appropriate total number of hierarchical levels in DLH should be achieved by overspecification, i.e. choosing the number of levels that would definitively be more than necessary for modelling the data. Overspecified DLH (in terms of levels) would then converge to the optimal structural complexity, as was demonstrated in Sec 4.3.
> > >
> > > At the same time, in Sec 4.3, we report another meaningful metric that is, perhaps, even more important in selecting the number of levels -- the total KL loss. In latent-variable generative models, latent KL divergence is often interpreted as a measure of the amount of information stored in the latent states. In DLH trained on the Moving Ball dataset, we observe an interesting property (Table 2): adding more hierarchical levels to a 2-level DLH does not produce statistically significant difference in the total value of the KL loss. We hypothesise that this happens precisely due to the overspecification of the model, i.e. its capacity is larger than necessary, and the model simply collapses the superfluous / unused latent levels. To demonstrate this more clearly, we added Table 5 in Appendix C.3, which shows the per-level values of the KL loss in DLH instances reported in Table 2. What you'll find is that in models with more than 2 levels, the KL divergence corresponding to levels >2 tends to the value of *zero*. We believe this is a critical indication of the fact that the model is overspecified for >2 levels, as the latent levels are not being used for modelling the data and no information is being stored in those levels. To make this absolutely clear, we have added interesting visualisations of this property in Figure 11 with explanations in Appendix C.3.
> > >
> > > At the same time, we can't help but recognise that the problem of choosing the number of levels is reminiscent of choosing the size of latent states in VAEs, or the number of layers in deep learning models. Although DLH can assist in providing indications of the optimal number of levels (tracking the average hierarchical depth and the total KL loss), the design choice nevertheless remains up to the human expert.
> > >
> > > ---
> > >
> > > **Q:** *What does \* indicate in Table 1?*
> > >
> > > **A:** Thank you for pointing this out -- we have now ammended the caption. The stars are meant to denote metrics with <5% standard deviation.
> > >
> > > ---
> > >
> > > **Q:** *In Figs 4-5, I strongly encourage the authors to include more qualitative comparisons for long-term video prediction with existing benchmarks. The proposed method is expected to generate future frames with higher visualization quality.*
> > >
> > > **A:** Absolutely. We have now added several qualitative comparisons in Appendix C.1.

---

> > > > ### Author Response · Authors · 2022-11-20
> > > > **Response to Reviewer 1qre (Part 4)**
> > > >
> > > > **Q:** *In addition, it would be nice if the authors could give clearer textual explanations in Figs 4-5. I am confused on what each image sequence represents...Are those in the first row the ground-truth images, or does each row represent a different prediction sequence by the model based on the same input sequence?*
> > > >
> > > > **A**: Thank you. We have now slightly editted the captions and added the necessary clarifications to the figures. For each dataset, the first row indicates the ground-truth sequence, while the bottom row shows DLH's open-loop prediction.
> > > >
> > > > ---
> > > >
> > > > **Q:** *If the experimental conditions allow, it is essential to show the model performance on more real-world datasets with large spatiotemporal uncertainty, such as RoboNet or BAIR. The authors may consider the action-conditioned video prediction setup because it would be interesting to analyze the correspondence between the learned indicators and the given action signals.*
> > > >
> > > > **A:** Thank you for this suggestion. We completely agree that it would be a great next step for exploring the potential of our model. However, it is unfortunately not possible for us to train DLH on more real-world datasets due to computational restrictions. Nevertheless, we plan to release the source code, including the scripts and instructions on how to train DLH for larger datasets, in order to allow  researchers with more resources to evaluate our model further. We believe that both RoboNet or BAIR are indeed very good candidates.
> > > >
> > > > With regards to action-conditioned datasets, this is definitely our next step, as one of our main motivations for working with hierarchical generative models for time-series is to apply this research to model-based reinforcement learning.
> > > >
> > > > ---
> > > >
> > > > In summary, we hope that we could address your main concerns regarding the novelty, qualitative comparisons, and the example in Figure 1. If so, we would appreciate if you could kindly reconsider your score. We would be very happy to hear back from you, if you require any further clarifications!

---

### Official Review · Reviewer_uT2U · 2022-10-24

**Confidence:** 3
**Correctness:** 3
**Technical Novelty And Significance:** 2
**Empirical Novelty And Significance:** 2
**Recommendation:** 5

**Clarity, Quality, Novelty And Reproducibility:**

* Clarity: The model is presented clearly.

* Quality: The experiments show that the model can improve upon previous hierarchical VAEs, but more work is needed to compare it to newer more performant methods on more complex datasets.

* Novelty: While the formulation is novel, the idea of hierarchical video VAEs has been extensively studied (see citations in the paper) and overall it is not clear that the more involved formulation presented in this paper will have practical impact.

* Reproducibility: The paper contains enough details to be reproduced.

**Strength And Weaknesses:**

**Strengths:**

[+] Clear method formulation and good empirical results on relatively simple datasets.


**Weaknesses:**

[-] The model is fairly similar to previous approaches that use hierarchical latent variables in time and/or space.
[-] Results are on relatively simple datasets, with newer video prediction methods being capable of generating sequences for long horizons on more complex datasets (e.g. Kinetics)


**Summary Of The Paper:**

The paper proposes a hierarchical VAE for video prediction. The latent structure of the model is formulated as a mixture of gaussians. The authors show that the proposed model is competitive to some previous hierarchical video VAE models on relatively simple datasets.

**Summary Of The Review:**

Overall the method presented in this paper is not that different from other previous hierarchical VAE models, and while it shows some improvements over these methods, in general it is unclear if this model will be of interest to the community compared to newer methods (FitVid, diffusion video models) that can generate long video sequences on complex datasets and have simpler latent spaces.

---

> ### Author Response · Authors · 2022-11-20
> **Response to Reviewer uT2U (Part 1)**
>
> Dear Reviewer uT2U,
>
> Thank you very much for your review of our paper. Please let us address some of the concerns you mentioned.
>
> **Q:** *The model is fairly similar to previous approaches that use hierarchical latent variables in time and/or space.*
>
> **A:** Thank you for this comment. Please allow us to briefly summarise why we believe the novel formulation of our approach holds significant weight.
>
> We would agree that our approach is conceptually similar to other hierarchical temporal VAEs that assume a separation of temporal scales over which latent variables evolve. However, we do not think that it is fair to discard the contributions of our work based on this argument. Indeed, we can imagine a number of potentially valuable and interesting latent modelling approaches that could fall under this class of models; we therefore do not believe this constitutes a valid argument against the novelty of any future work in this area.
>
> With respect to DLH, to our knowledge, this work is the first work to formalise the temporal evolution of states in hierarchical-temporal VAEs using Gaussian mixtures. This result is important, as it allowed for a theoretically-sound non-parametric estimation of the hierarchical indicator variables, which is inspired by the relevant literature on GM-VAEs. We strongly believe this contribution is substantial, as we bypass the problematic parametric estimation of discrete latent variables (e.g. in VTA), which are also known to scale poorly in hierarchical models. The benefits of our approach are similarly substantiated by the results presented in Section 4, including significant improvements over the baseline models.
>
> We sincerely hope you reconsider your opinion on this matter. Please let us know if you require any further clarifications -- we would be very happy to hear back from you.
>
> ---
>
> **Q:** *Results are on relatively simple datasets, with newer video prediction methods being capable of generating sequences for long horizons on more complex datasets (e.g. Kinetics)*
>
>
> **Q:** *The experiments show that the model can improve upon previous hierarchical VAEs, but more work is needed to compare it to newer more performant methods on more complex datasets.*
>
> **A:** Thank you for these comments -- we appreciate your concern. However, we believe it is important to take into account the fact that newer methods require a substantial amount of computational resources. For example, the FitVid model you mentioned contains >300M parameters, compared to only 7M parameters in DLH; (CW-VAE 12M, LMC-Memory 34M, VTA 3M). With regards to using more complex datasets, we certainly agree that this would be very useful; however, more complex datasets similarly require more compute. As much as we would wish to make such comparisons, as well to scale DLH itself, we are simply limited in our computational resources to do so. Nevertheless, we believe that our approach was  appropriately compared to other VAE-based models (with similar parameter counts) using well-established datasets, and would be of significant interest to the community. We are also planning to release the code, which would make it possible for other researchers with more resources to scale the model.
>
> ---
>
> **Q:** *While the formulation is novel, the idea of hierarchical video VAEs has been extensively studied (see citations in the paper) and overall it is not clear that the more involved formulation presented in this paper will have practical impact.*
>
> **A:** We appreciate that you recognise the novelty of our formulation. We believe that the practical impact of our paper has been substantiated by the number of empirical results presented in Section 4. These included: improved long-horizon video prediction against baseline models, hierarchical spatiotemporal disentanglement resulting from the formulation of the latent states as Gaussian mixtures, natural ability to better deal with temporal stochasticity, and the dynamic simplification of DLH's structural complexity depending on the dataset. We would be delighted if you could clarify this point a bit more, or if we can do anything to further delineate the value and novelty of our approach.

---

> > ### Author Response · Authors · 2022-11-20
> > **Response to Reviewer uT2U (Part 2)**
> >
> > **Q:** *in general it is unclear if this model will be of interest to the community compared to newer methods (FitVid, diffusion video models) that can generate long video sequences on complex datasets and have simpler latent spaces.*
> >
> > **A:** To your point about simpler latent spaces, we would like to stress that we do not believe that the latent space of DLH is complicated. It is a modular formalisation of the hierarchical temporal VAEs with latent variables operating over jumpy timescales. Furthermore, in the paper, we show how, in virtue of our design of the latent space, we arrive at the important properties of hierarchical spatiotemporal disentanglement. As mentioned in the discussion section, we believe that increasing the focus of video prediction models on representation learning (including via the design of more intricate latent states) evidently provides significant benefits to the model's predictive capabilities. In our opinion, this is not a disadvantage by any means. We would be very happy to discuss this point further.
> >
> > ---
> >
> > Thank you once again for your review. We hope we could adress some of your concerns, and we would be very happy to hear back from you.

---

### Official Review · Reviewer_2ayA · 2022-10-24

**Confidence:** 3
**Correctness:** 4
**Technical Novelty And Significance:** 3
**Empirical Novelty And Significance:** 3
**Recommendation:** 6

**Clarity, Quality, Novelty And Reproducibility:**

The overall presentation and quality of the paper was quite high. Although the method builds on components from existing approaches, the dynamic hierarchical architecture for video prediction appears to be novel. Aside from the points of clarification listed above, I found the paper to be clear. There does not appear to be code provided in the supplemental material, which hurts reproducibility.

**Strength And Weaknesses:**

Strengths:
* I really enjoyed reading this paper. It is well written and the ideas are easy to follow.
* The problem is well-motivated and the literature review does a good job at contextualizing the paper in prior work.
* The methods section was presented with intuition for the design choices and notation. This intuition was then supported empirically.
* An attempt was made to interpret the hierarchical levels semantically in the experiments.
* Given the considered datasets, the experiments section is constructed thoughtfully and the results are convincing.
* The figures are informative and effectively illustrate the benefits of the proposed approach. Particularly, the intuition provided by the results in Fig. 5 was helpful in understanding how the method works.

Weaknesses:
* Overall, the datasets considered are fairly uncluttered and simplistic. The video examples do not highlight the capabilities of DLH to handle many objects at many different speeds (e.g., in crowded urban scenes). The data also does not showcase what happens when the background is not stationary and there are moving objects. I would recommend considering a more complicated, dynamic dataset, for example, from the autonomous driving setting (i.e., Waymo Open Dataset [1], NuScenes [2], or KITTI [3]). The simplicity of the toy Moving Ball dataset is also underscored by the results of Table 2, where the full capacity of the hierarchical model is not necessary to model the data. Although it is great to see that the model can dynamically adapt to use less of the latent space when the underlying data distribution is simpler, it would be compelling to see how the full latent space would be used in a more complex setting.
* Although there is a discussion on the importance of stochasticity, this capability is only explored in a toy-dataset setting with random color changes. No discussion of multimodality in the distribution is included. Could DHL handle multimodal outputs (e.g., multiple equally valid possibilities for the future)? This would again be relevant to more complex video datasets (e.g., in the urban setting), where given observations of a person walking straight, they could choose to continue walking straight or turn in the future.
* It would be helpful in Fig. 4 or in an appendix to show the outputs of the baseline approaches for comparison.
* In Fig. 5, it is not entirely clear that levels 1 and 2 for the KTH Action dataset are disentangled.
* It would be interesting to benchmark against a deterministic video prediction method in Table 1 to see if the considered datasets are sufficiently stochastic to warrant modeling of stochasticity.
* Is there a way to report a measure of statistic significance of the proposed method's metric performance over the baselines in Table 1?
* In Fig. 6, it is not very clear to me what is wrong with some of the highlighted frames output by the CW-VAE. Is the issue that the ball reduces in size for those frames? Why were the other baseline results not shown?
* Further explanation for the values in Table 3 would be helpful.

[1] Sun, Pei, et al. "Scalability in perception for autonomous driving: Waymo open dataset." CVPR, 2020.

[2] Caesar, Holger, et al. "nuScenes: A multimodal dataset for autonomous driving." CVPR, 2020.

[3] Geiger, Andreas, et al. "Vision meets robotics: The KITTI dataset." IJRR, 2013.

Some typos and minor points of confusion are listed below:

1. I am not sure I fully followed the diagrams in Fig. 2. Are there temporal indices missing from the states?
2. Are there hierarchical levels and $\psi$ parameters missing in the 'Estimating' paragraphs in Sec. 2.2 and similarly dropped indexes in $p(e \mid s)$ in the paragraph before Eq. 3?
3. In Eq. 3, $\Lambda$ is not defined.
4. Missing period at the end of Eq. 3.
5. In Sec. 2.3, I am having some trouble understanding the notation. Should $q(\bf{e}^{n+1} \mid \bf{e}^n = 0) = 0$ be $q(\bf{e}^{n+1} = 1 \mid \bf{e}^n = 0) = 0$?
6. It would be helpful to derive Eq. 8 from Eq. 7 in the appendix for completeness.
7. Fig. 3 was referenced much earlier than it appears.
8. Am I correct in understanding that at the first hierarchical level $e^1 = 1$ always?
9. In Fig. 4, it should be made clear whether the 30 past context frames are included in the visualization or only the 100 predicted frames are shown.
10. In Sec. 3, temporal abstraction paragraph, the sentence "Temporal abstraction models ..." has a grammatical typo, is a bit long, and is missing a period at the end.
11. Missing periods at the end of table captions. Unclear what the * symbol signifies.
12. In Sec. 4.3, I did not fully understand what it means that "DLH learns transition between progressively slower features in the higher levels of its hierarchy". Does this mean that minor variations in the scene are faster features than location changes of the view?
13. The references should be proofread (e.g., to ensure the year is not entered twice in a citation, the conference venue is listed instead of ArXiv when available, the confererence name formatting is consistent, etc.).

**Summary Of The Paper:**

The paper presents a method for hierarchical representation learning of spatiotemporal features in long-term video prediction. The proposed method is called: Dynamic Latent Hierarchy (DLH). The method distinguishes between features that are changing and those that are not changing in the video sequence. DLH is able to handle multiple objects moving at different speeds and differentiate moving objects from a static environment. The advantages of DLH include: long-term video prediction, improved modeling of stochasticity, and dynamic, efficient latent structure. DLH outperforms baseline approaches on Moving MNIST, KTH Action, and DML Mazes datasets.

**Summary Of The Review:**

Overall, this is a good, clearly written paper that proposes a reasonable approach for video prediction that outperforms the considered baselines across several datasets. I am currently inclined to accept it. I encourage the authors address the first two weaknesses listed above, in particular, regarding the complexity of the datasets used for evaluation.

---

> ### Author Response · Authors · 2022-11-20
> **Response to Reviewer 2ayA (Part 1)**
>
> Dear Reviewer 2ayA,
>
> Thank you for such a detailed and insightful review of our paper! In what follows, we wish to address some of the questions you raised.
>
> **Q:** *Overall, the datasets considered are fairly uncluttered and simplistic. The video examples do not highlight the capabilities of DLH to handle many objects at many different speeds (e.g., in crowded urban scenes). The data also does not showcase what happens when the background is not stationary and there are moving objects. I would recommend considering a more complicated, dynamic dataset, for example, from the autonomous driving setting (i.e., Waymo Open Dataset [1], NuScenes [2], or KITTI [3]). The simplicity of the toy Moving Ball dataset is also underscored by the results of Table 2, where the full capacity of the hierarchical model is not necessary to model the data. Although it is great to see that the model can dynamically adapt to use less of the latent space when the underlying data distribution is simpler, it would be compelling to see how the full latent space would be used in a more complex setting.*
>
> **A:** Thank you for your recommendation. This is definitely an aspect of our work which was challenging. Being able to train our model with more complex datasets would undoubtedly make the presentation of our paper more impressive, and our task of evaluating the capabilities of our model more straightforward. However, even though our approach is shown to achieve superior perfrormance against the baseline methods, we unfortunately do not possess the necessary computational resources for scaling up further.
>
> Nevertheless, we believe that we selected a number of well-established datasets, which were also used in the testing of the baseline methods. We therefore think that the datasets are appropriate to showcase the improved predictive capability of our model.
>
> Furthermore, the selection of the Moving Ball dataset was motivated by (1) its simplicity and (2) the ease of interpretability. The first factor allowed us to showcase the DLH's ability to simplify its hierarchical structure, while the second factor was important for the stochasticity results in Section 4.4. With regards to the other datasets, we believe that the video prediction results showcase the effectiveness of DLH employing the 'full latent space'.
>
> Finally, to increase transparency and allow other researchers with more resources to easily evaluate our method further, we are planning to release all our source code after the review process.

---

> > ### Author Response · Authors · 2022-11-20
> > **Response to Reviewer 2ayA (Part 2)**
> >
> > ---
> >
> > **Q:** *Although there is a discussion on the importance of stochasticity, this capability is only explored in a toy-dataset setting with random color changes. No discussion of multimodality in the distribution is included. Could DHL handle multimodal outputs (e.g., multiple equally valid possibilities for the future)? This would again be relevant to more complex video datasets (e.g., in the urban setting), where given observations of a person walking straight, they could choose to continue walking straight or turn in the future.*
> >
> > **A:** This is a great question, as it nicely relates to an important direction of future work  that naturally follows from the theoretical formulation of our model using the temporal mixture of Gaussians. In particular, this formulation enables us to extend our model to incorporate multiple predictions of the future by means of increasing the number of components in an MoG, such that there is now multiple *change* priors. We want to emphasise this again, as it relates to the novelty of our work: because of the MoG formulation, this extension does not require major changes in the inference procedure, as the same non-parametric inference techniques can be applied.
> >
> > With regards to the current version of DLH and the stochasticity analysis, we believe that the presented results showcase a valuable property, despite the remaining limitations with respect to multimodal predicted outputs. To see this, we can break down the problem of predicting the next stochastic realisation of the world into two levels of difficulty: (1) the state has remained the same or the state has changed, and (2) the state has remained the same or the state has changed into one of the possible future realisations (i.e. multimodal). We believe that these two levels relate to the amount of expressivity a model must possess in its definition of the latent states. Classical temporal VAEs (e.g. CW-VAE) cannot deal with either of these two expressivity levels, as their predicted states must contain information about all of the possible outcomes (including the *static* state) and their implicit probabilities of occurring. In contrast, DLH takes a step forward and effectively deals with the first level of this problem by increasing the expressivity of the latent state by means of explicitly separating the two possible outcomes within the temporal mixture of Gaussians and learning the probability of their occurrence. Although DLH does not at this point deal with the second level, it nevertheless offers the natural basis for extending the model's expressivity to capture multiple future outcomes. As mentioned, we can apply the same methodology for the case where multiple *change* priors are predicted, thus effectively scaling the model for more complex stochastic dynamics. At this time, we leave it for future work.
> >
> > ---
> >
> > **Q:** *It would be helpful in Fig. 4 or in an appendix to show the outputs of the baseline approaches for comparison.*
> >
> > **A:** Great suggestion. We have now added prediction comparisons against other models in Appendix C.1.
> >
> > ---
> >
> > **Q:** *In Fig. 5, it is not entirely clear that levels 1 and 2 for the KTH Action dataset are disentangled.*
> >
> > **A:** Thank you for this comment -- it is a good observation. We do not believe that levels 1 and 2 are entirely disentangled either. The results indicate that they are indeed both encoding motion, even though it seems more pronounced in level 2. However, we believe that these results are very much in line with what one might expect to see in a DLH model training on KTH. In particular, some levels may exchange information by means of hierarchical dependencies. For the KTH dataset, it actually makes sense to encode motion in levels 1 and 2, but not in level 3. This is because, in KTH, motion is a continuously changing feature with periodic pauses (e.g. person freezing their pose for a brief second). Indeed, in Figure 5c we show exactly that -- while level 1 is forced to update at every timestep (via the assumption in Sec 2.3), level 2 updates only when the person's arms are in motion. In this case, level 2 should contain more detailed representations of the periodic motions, but this does not restrict level 1 from representing motion-related features either; on the contrary, the two levels can 'work together' using the hierarchical relationship to represent a person's pose. Evidently, this is the property we observe in Figure 5a.

---

> > > ### Author Response · Authors · 2022-11-20
> > > **Response to Reviewer 2ayA (Part 3)**
> > >
> > > ---
> > >
> > > **Q:** *It would be interesting to benchmark against a deterministic video prediction method in Table 1 to see if the considered datasets are sufficiently stochastic to warrant modeling of stochasticity.*
> > >
> > > **A:** We really like this suggestion; however, regretfully, we did not have enough time to implement this idea.
> > >
> > > ---
> > >
> > > **Q:** *Is there a way to report a measure of statistic significance of the proposed method's metric performance over the baselines in Table 1?*
> > >
> > > **A:** In writing this paper, we made sure that our model outperforms the baseline approaches with a reasonable degree of statistical significance. We can therefore confirm that the presented results are statistically significant. Unfortunately, there was not enough space to include a detailed analysis of this in the paper. However, in line with the standard way of presenting such results in video prediction literature, we have added a useful indication of the standard deviation for each metric (*).
> > >
> > > ---
> > >
> > > **Q:** *In Fig. 6, it is not very clear to me what is wrong with some of the highlighted frames output by the CW-VAE. Is the issue that the ball reduces in size for those frames? Why were the other baseline results not shown?*
> > >
> > > **A:** The problem with these frames is the apparent quality degeneration; e.g. CW-VAE produces colours that do not exist in the dataset, deforms the ball size, etc. We believe this is largely caused by the dataset's temporal stochasticity and the CW-VAE's inability to properly represent it. Due to the space constraints, we thought it would be sufficient to present the predictions of CW-VAE as a representative hierarchical temporal VAE for video prediction.
> > >
> > > ---
> > >
> > > **Q:** *Further explanation for the values in Table 3 would be helpful.*
> > >
> > > **A:**  Thank you for this suggestion. We have now added further clarififications in Section 4.
> > >
> > > Recall that the prior component model $p(e^n_t|\cdot)$ is meant to predict hierarchical state changes for level $n$ and timestep $t$. However, as the Moving Ball dataset becomes more stochastic, the timesteps at which these changes actually occur become more difficult to predict. We are therefore showing the average predicted probabilities of $p_\theta(e_t=1|\cdot)$, given an inferred posterior component being change ($1 \sim q(e_t)$) or static ($0 \sim q(e_t)$). For the deterministic Moving Ball, the prior component model should learn to perfectly mimic the state changes inferred by the posterior component model. In contrast, as the stochasticity of the dataset rises, it becomes increasingly difficult for the prior model to predict the exact timesteps at which state changes are inferred by the posterior model; the model should thus become more cautious in its predictions. This is precisely what we can observe in Table 3.
> > >
> > > ---
> > >
> > > **Some typos and minor points of confusion are listed below:**
> > >
> > > ---
> > >
> > > **Q:** *I am not sure I fully followed the diagrams in Fig. 2. Are there temporal indices missing from the states?*
> > >
> > > **A:** Indeed, the temporal indices were missing -- this has now been fixed. Thank you.
> > >
> > > ---
> > >
> > > **Q:** *Are there hierarchical levels and parameters missing in the 'Estimating' paragraphs in Sec. 2.2 and similarly dropped indexes in the paragraph before Eq. 3?*
> > >
> > > **A:** In certain places of the paper, we wanted to simplify the notation to make it easier to read; however, this may have caused some confusion instead. We have now updated all the relevant notation to explicitly state the conditional dependencies and parametrisations.
> > >
> > > ---
> > >
> > > **Q:** *In Eq. 3, lambda is not defined.*
> > >
> > > **A:** Lambda is simply the likelihood ratio between the competing prior models. In Appendix B.1, we have now added a detailed derivation of the expected log-likelihood ratio.
> > >
> > > ---
> > >
> > > **Q:** *Missing period at the end of Eq. 3.*
> > >
> > > **A:** Fixed!
> > >
> > > ---
> > >
> > > **Q:** *In Sec. 2.3, I am having some trouble understanding the notation. Should $q(e=1|e=0)=0$ be?*
> > >
> > > **A:** Absolutely, thank you for pointing this out.
> > >
> > >
> > > ---
> > >
> > > **Q:** *Am I correct in understanding that at the first hierarchical level $e^1=1$ always?*
> > >
> > > **A:** This is correct.
> > >
> > > ---
> > >
> > > **Q:** *In Fig. 4, it should be made clear whether the 30 past context frames are included in the visualization or only the 100 predicted frames are shown.*
> > >
> > > **A:** Thank you for this comment. We have now slightly editted the caption to include this information. The context frames are not included in the visualisation.
> > >
> > > ---
> > >
> > > **Q:** *In Sec. 3, temporal abstraction paragraph, the sentence "Temporal abstraction models ..." has a grammatical typo, is a bit long, and is missing a period at the end.*
> > >
> > > **A:** Thank you -- this has now been fixed.
> > >
> > > ---
> > >
> > > **Q:** *Missing periods at the end of table captions. Unclear what the \* symbol signifies*.
> > >
> > > **A:** All fixed!

---

> > > > ### Author Response · Authors · 2022-11-20
> > > > **Response to Reviewer 2ayA (Part 4)**
> > > >
> > > > **Q:** *In Sec. 4.3, I did not fully understand what it means that "DLH learns transition between progressively slower features in the higher levels of its hierarchy". Does this mean that minor variations in the scene are faster features than location changes of the view?*
> > > >
> > > > **A:** By the design of DLH's generative structure, the higher levels of the model update over slower timescales. This means that they are encouraged to represent slower evolving features of the scene, and by extension learn to transition these features. As such, minor variations in the scene (such as view angle) are expected to be represented in the lower levels, while more general contextual information (location, maze characteristics) in the higher levels. We observe this exact effect in Figure 5b, and this is what is being described in the paragraph in Section 4.3.
> > > >
> > > > ---
> > > >
> > > > **Q:** *The references should be proofread (e.g., to ensure the year is not entered twice in a citation, the conference venue is listed instead of ArXiv when available, the confererence name formatting is consistent, etc.).*
> > > >
> > > > **A:** Thank you for this. We will make sure to proofread all of the citations in the final draft of the paper, in the case it is accepted.
> > > >
> > > > ---
> > > >
> > > > Thank you once again for your detailed review. We really appreciate it. Please let us know if we could address your concerns, or if you have any further questions about the paper.

---

### Author Response · Authors · 2022-11-20
**To all Reviewers**

Dear Reviewers,

Thank you so much for your feedback on our paper. We found your comments and suggestions to be both helpful and thought-provoking. We have now uploaded the revised version of the paper, in which we attempted to accommodate most of your requests. We have referenced any changes made to the paper in the submitted responses.

As the authors of the paper, we also appreciate that you found our paper to be well-motivated and clearly written, as well as the results to be convincing.

We sincerely hope that we could address most of your concerns and would be happy to have further discussion on any remaining questions you may have about the paper. If you feel like your questions have been answered, we would appreciate if you could consider increasing the scores. Let us know if your require further clarifications on any of our responses.

Kind regards,
the Authors

---

### Decision · Program_Chairs · 2023-01-20

**Decision:**

Reject

**Justification For Why Not Higher Score:**

- The paper is not very novel compared to other hierarchical VAE and jumpy state transition approaches and its contribution is marginal, especially given recent development of video models and that the results are not that strong.
- Some of the claims of the paper are not well supported.
- Writing can be improved both of the model and the experiments.

**Justification For Why Not Lower Score:**

N/A

**Metareview: Summary, Strengths And Weaknesses:**

This paper presents deep hierarchical latent model of video, where different layers update on different time scales.
- The paper is not very novel compared to other hierarchical VAE and jumpy state transition approaches and its contribution is marginal, especially given recent development of video models and that the results are not that strong.
- Some of the claims of the paper are not well supported.
- Writing can be improved both of the model and the experiments.